# CURVATURE EXPLAINS LOSS OF PLASTICITY

## ABSTRACT

Loss of plasticity is a phenomenon in which neural networks lose their ability to learn from new experience. Despite being empirically observed in several problem settings, little is understood about the mechanisms that lead to loss of plasticity. In this paper, we offer a consistent explanation for plasticity loss, based on an assertion that neural networks lose directions of curvature during training and that plasticity loss can be attributed to this reduction in curvature. To support such a claim, we provide a systematic empirical investigation of plasticity loss across several continual supervised learning problems. Our findings illustrate that curvature loss coincides with and sometimes precedes plasticity loss, while also showing that previous explanations are insufficient to explain loss of plasticity in all settings. Lastly, we show that regularizers which mitigate loss of plasticity also preserve curvature, motivating a simple distributional regularizer that proves to be effective across the problem settings considered.

## 1 INTRODUCTION

A longstanding goal of machine learning research is to develop algorithms that can learn *continually* and cope with unforeseen changes in their environment (Sutton et al., 2007). Current learning algorithms, however, struggle to learn from dynamically changing targets and are unable to adapt gracefully to unforeseen environment changes during the learning process (Abbas et al., 2023, Dohare et al., 2023a, Lyle et al., 2023, Zilly et al., 2021). Such limitations can be seen to be a byproduct of following a supervised learning paradigm that assumes the problem is stationary. Recently, there has been growing recognition of the fact that there are limitations to what can be learned from a fixed and unchanging dataset (Hoffmann et al., 2022) and that there are implicit non-stationarities in many learning problems of interest (Igl et al., 2021).

The concept of plasticity has been receiving growing attention in the continual learning literature, where the loss of plasticity—a reduction in the ability to learn new things (Dohare et al., 2023a, Lyle et al., 2023)—has been noted as a critical shortcoming in current learning algorithms. That is, learning algorithms that are performant in the non-continual learning setting often struggle when applied to continual learning problems, exhibiting a striking loss of plasticity such that learning slows down or even halts after successive changes in the learning environment. Such a loss of plasticity can be readily observed in settings where a neural network must continue to learn after changes occur in the observations or targets.

Several aspects of a learning algorithm and modeling architecture have been found to contribute to, or mitigate, loss of plasticity, such as the optimizer (Dohare et al., 2023a), the step-size (Ash & Adams, 2020, Berariu et al., 2021), the number of updates (Lyle et al., 2023), the activation function (Abbas et al., 2023) and the use of specific regularizers (Dohare et al., 2021, Kumar et al., 2023, Lyle et al., 2021). Such factors hint that there might be simpler underlying mechanisms that are the root cause of the loss of plasticity. For example, the success of several methods that regularize the neural network towards properties of the initialization suggests that some property of the initialization mitigates loss of plasticity. Unfortunately, no such property has yet been identified. Some factors that have been found to correlate with loss of plasticity include, a decrease in the gradient norm (Abbas et al., 2023), a reduction in the rank of the learned representations weight tensors (Gulcehre et al., 2022, Kumar et al., 2021), neuron dormancy (Sokar et al., 2023), and an increase in the norm of the parameters (Nikishin et al., 2022).

In this paper, we propose that loss of plasticity can be explained by a loss of curvature. Our work contributes to a growing literature on the importance of curvature for understanding neural network dynamics (Cohen et al., 2021, Fort & Ganguli, 2019, Hochreiter & Schmidhuber, 1997). Within the continual learning and plasticity literature, the assertion that curvature is related to plasticity is relatively new (Lyle et al., 2023). In contrast to the general assertion that curvature is related to plasticity, our work specifically posits that loss of curvature explains loss of plasticity. In particular, we provide empirical evidence that supports a claim that loss of plasticity co-occurs with a reduction in the rank of the Hessian of the training objective at the beginning of a new task.

This work improves the understanding of plasticity loss in continual supervised learning by,

1. systematically outlining previous explanations for loss of plasticity, and providing counter-examples that illustrate conditions where these explanations fail to account for particular increases or decreases in plasticity;

2. positing that loss of curvature, as measured as the change in the rank of the Hessian of the training objective, is the underlying cause of loss of plasticity and demonstrating that loss of curvature coincides with loss of plasticity across several factors and benchmarks;

3. proposing a regularizer that keeps the distribution of weights close to the initialization distribution, and showing that this allows the parameters to move further from initialization while preserving curvature for successive tasks.

## 2 Contributing Factors and Explanations for Plasticity Loss

Before defining what we mean by loss of plasticity, we will outline the continual supervised learning problem setting that we study. We assume the learning algorithm operates in an on-line manner, processing an observation-target pair $(x, y)$ and updating the neural network parameters $\theta$ after each pair. In continual supervised learning, there is a periodic and regular change every $U$ updates to the distribution generating the observations or targets. For every $U$ updates, the neural network must minimize an objective defined over a new distribution and we refer to this new distribution as a *task*. The problem setting is designed so that the task at any point in time has the same difficulty.[1] We are primarily interested in error at the the end of task $K$ averaged across all observations in that task, $J_K = J(\theta_{UK}) = \mathbb{E}_{p_k}(\ell(f_{\theta_{UK}}(x), y))$ for some loss function $\ell$.

Although loss of plasticity is an empirically observed phenomenon, the way it is measured in the literature can vary. In this paper, we will use loss of plasticity to refer to the phenomenon that $J_K$ increases rather than decreases as a function of $K$. Some works evaluate learning and plasticity with the average online error over the learning trajectory within a task (Dohare et al., 2023a, Elsayed & Mahmood, 2023, Kumar et al., 2023). While the two are related, we focus on the error at the end of the task to remove the effect of increasing error at the beginning of a subsequent task, which can suggest loss of plasticity in the online error even if the error at the end of a task is constant (see Appendix C.4). Because the error increases as more tasks are seen, this means that the neural network is struggling to learn from the new experience given by the subsequent task.

### 2.1 Factors That Can Contribute to Loss of Plasticity

Given a concrete notion of plasticity, we reiterate that the underlying mechanisms leading to loss of plasticity have been so-far elusive. This is partly because multiple factors can contribute to loss of plasticity, or even mitigate it. In this section, we summarize some of these potential factors before surveying previous explanations for the underlying mechanism behind loss of plasticity.

**Optimizer** Optimizers that were designed and tuned for stationary distributions can exacerbate loss of plasticity in non-stationary settings. For instance, the work by Lyle et al. (2023) showed empirically that Adam (Kingma & Ba, 2015) can be unstable on a subsequent task due to its momentum and scaling from a previous task.

**Step-size** In addition to the optimizer, an overlooked fact is that the step-size itself is a crucial factor in both contributing to and mitigating the loss of plasticity. The study by Berariu et al. (2021), for

---

[1] A suitably initialized neural network should be able to equally minimize the objective for any of the tasks we consider.

example, suggests that the plasticity loss is preventable by amplifying the randomness of gradients with a larger step-size. These findings extend to other hyper-parameters of the optimizer: Properly tuned hyper-parameters for Adam, for example, can mitigate loss of plasticity leading to policy collapse in reinforcement learning (Dohare et al., 2023b, Lyle et al., 2023).

**Update Budget** Continual supervised learning experiments, including those below, use a fixed number of update steps per task (e.g., Abbas et al., 2023, Elsayed & Mahmood, 2023, Javed & White, 2019). Despite the fact that the individual tasks themselves are of the same difficulty, the neural network might not be able to escape its task-specific initialization within the pre-determined update budget. Lyle et al. (2023) show that, as the number of update steps increase in a first task, learning slows down on a subsequent task, requiring even more update steps on the subsequent task to reach the same training error.

**Activation function** One major factor that can contribute or mitigate loss of plasticity is the activation function. Work by Abbas et al. (2023) suggests that, in the reinforcement learning setting, loss of plasticity occurs because of an increasing portion of hidden units being set to zero by `ReLU` activations (Fukushima, 1975, Nair & Hinton, 2010). The authors then show that `CReLU` (Shang et al., 2016) prevents saturation, mitigating plasticity loss almost entirely. However, other works have shown that plasticity loss can still occur with non-saturating activation functions (Dohare et al., 2021, 2023a), such as `leaky-ReLU` (Xu et al., 2015).

**Properties of the objective function and regularizer** The objective function being optimized greatly influences the optimization landscape and, hence, plasticity (Lyle et al., 2021, 2023). Regularization is one modification to the objective function that helps mitigate loss of plasticity. When L2 regularization is properly tuned, for example, it can help mitigate loss of plasticity (Dohare et al., 2023a). Another regularizer that mitigates loss of plasticity is regenerative regularization ,which regularizes towards the parameter initialization (Kumar et al., 2023).

## 2.2 Previous Explanations for Loss of Plasticity

Not only are there several factors that could possibly contribute to loss of plasticity, there are also several explanations for this phenomenon. We survey the recent explanations of loss of plasticity below. In the next section, we present results showing that none of these explanations are sufficient to explain loss of plasticity across different problem settings.

**Decreasing update/gradient norm** The simplest explanation for loss of plasticity is that the update norm goes to zero. This would mean that the parameters of the neural network stop changing, eliminating all plasticity. This tends to occur with a decrease in the norm of the features for particular layers (Abbas et al., 2023, Nikishin et al., 2022).

**Dormant Neurons** Another explanation for plasticity loss is a steady increase in the number of inactive neurons, namely, the dormant neuron phenomenon (Sokar et al., 2023). It is hypothesized that fewer active neurons decreases a neural network's expressivity, leading to loss of plasticity.

**Decreasing representation rank** Related to the effective capacity of a neural network, lower representation rank suggests that fewer features are being represented by the neural network (Kumar et al., 2021). It has been observed that decreasing representation rank is sometimes correlated with loss of plasticity (Dohare et al., 2023a, Kumar et al., 2023, Lyle et al., 2023). With a similar intuition as the representation rank, a decreasing rank of the weight matrices may prevent a neural network from representing certain features and hence lower plasticity (Gulcehre et al., 2022, Lyle et al., 2023).

**Increasing parameter norm** An increasing parameter norm is sometimes associated with loss of plasticity in both continual and reinforcement learning (Dohare et al., 2023a, Nikishin et al., 2022), but it is not necessarily a cause (Lyle et al., 2023). The reason for parameter norms to increase and lead to loss of plasticity is not clear, perhaps suggesting a slow divergence in the training dynamics.

## 3 Empirical Counter-examples of Previous Explanations

As a first step, we investigate the factors influencing and explanations of plasticity described in Section 2. The goal of this section is primarily to provide counterexamples to the different explanations

of loss of plasticity, showing that the existing explanations fail to fully explain the phenomenon. To do so, we use the MNIST dataset (LeCun et al., 2010) where the labels of each image are periodically shuffled. While MNIST is a simple classification problem, label shuffling highlights the difficulties associated with maintaining plasticity and was previously demonstrated as such (Kumar et al., 2023, Lyle et al., 2023). In this section, we focus on this MNIST task for its simplicity, showing that even in a simple setting, one can find counter-examples to previous explanations in the literature. We emphasize that the goal here is merely to uncover simple counter-examples that refute proposed explanations for loss of plasticity, not investigate the phenomenon more broadly. In Section 6, we extend our investigation to the other common benchmarks for loss of plasticity in continual supervised learning.

**Methods**   In this experiment, we vary only the activation function between `ReLU`, `leaky-ReLU`, `tanh` and the `identity`. Previous work found that the activation function has a significant effect on the plasticity of the neural network. We train a neural network for 200 epochs per task with a total of 100 tasks and measure the error across all observations at the end of each task.

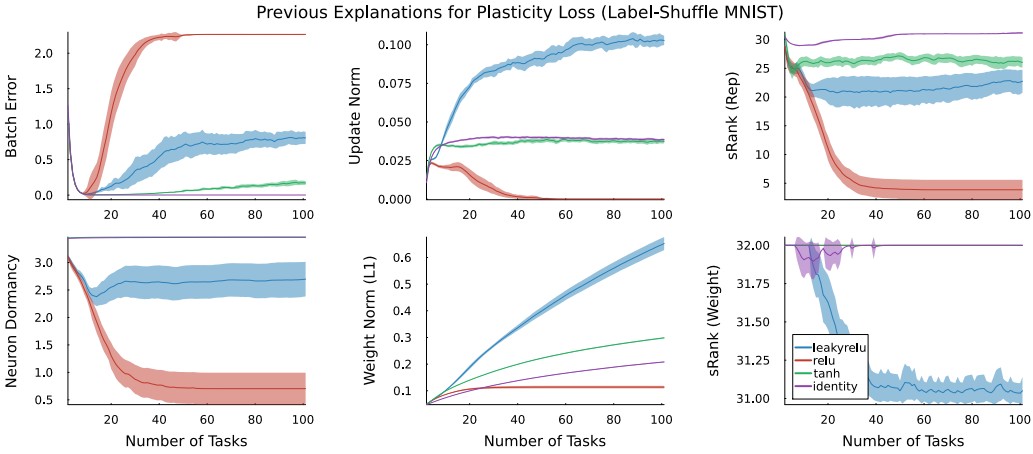

Figure 1: All results have a shaded region corresponding to a 95% confidence interval of the mean. Top left: All non-linear activation functions lose plasticity, but a deep linear network with identity activations does not. Rest: None of the aforementioned explanations explain the difference between plasticity loss of different activation functions. For neuron dormancy and the weight stable rank, some activation functions overlap and are not visible.

**Results**   The main result of this experiment can be found in Figure 1. Our findings show that none of the aforementioned explanations of loss of plasticity explain the phenomenon across different activation functions in this simple problem. In the top-left figure, we have the task-end batch training error as a function of the number of tasks. All non-linear activation functions can achieve low error on the first few tasks, but this error increases over time. The deep linear network (a neural network with an `identity` activation function) is able to maintain a low training error for each of the 100 tasks. This suggests that loss of plasticity only occurs with the non-linear activations, and that `ReLU` loses its plasticity the quickest.

The remaining plots in Figure 1 show the measurement of quantities pertaining to the aforementioned explanations of plasticity loss. A decreasing update norm, for example, may seem like an intuitive explanation of loss of plasticity. However, in the top-middle plot, we see that the update norm consistently increases for the `leaky-ReLU` activation function, making the explanation inconsistent. A similar inconsistency in the rank of the representation makes that explanation inconsistent with loss of plasticity, too. The same is true for neuron dormancy (measured as the entropy of the activations), weight norm and weight rank. There exists at least one activation such that the trend in the training error does not agree with the trend in the explanation (see Appendix A for the further analysis).

# 4 MEASURING CURVATURE OF A CHANGING OPTIMIZATION LANDSCAPE

One missing piece in the explanations previously proposed is the curvature of the optimization landscape. While previous work pointed out that curvature is connected to plasticity (Lyle et al., 2023), our work specifically posits that loss of curvature coincides with and, in some cases, precedes plasticity loss. Our experiments in Section 6 show that plasticity loss occurs when, at the beginning of each new task, the optimization landscape has diminishing curvature.

Before presenting empirical evidence of the relationship between plasticity and curvature, we briefly describe the effect that task and data changes have on the curvature of the optimization landscape. The local curvature of the optimization landscape at a particular parameter $\theta$, is expressed by the Hessian of the objective function, $H_t(\theta) = \nabla_\theta^2 J_t(\theta)\big|_{\theta=\theta_t}$. We hide the dependence on data in the training objective and the Hessian, and instead index both by time. For conventional supervised learning problems, the training objective is stationary because the input and target distributions do not change. For non-stationary learning problems, like those in continual learning, the distributions underlying the observations and targets will change over time. Thus there can be changes in the objective, gradient and Hessian that is due to the data changing and not due to the parameters.

We are interested in how the curvature of the optimization landscape changes when the task changes. Of particular interest is the rank of the Hessian after a task change because, if it is decreasing, then there are fewer directions of curvature to learn on the new task. For simplicity, and in alignment with our experiments, we will assume that each task has an update budget $U$. Then the training objective on the $K$-th task will be stationary for $U$ steps. When the task changes, at $t = UK + 1$, the Hessian will change due to changes in the data - and not due to changes in the parameters. We measure the rank at the beginning of the task by the *stable rank*, srank$(H_{UK+1}(\theta))$, where srank$(M) = \min\left\{ j : \frac{\sum_{i=1}^j \sigma_i(M)}{\sum_{i=1}^d \sigma_i(M)} > 0.99 \right\}$ is the stable rank and $\{\sigma_i(M)\}_{i=1}^d$ are the singular values arranged in decreasing order. The stable rank specifies the number of basis vectors needed to represent most of image of the matrix $M$ (Kumar et al., 2021, Yang et al., 2019).

## 4.1 PARTIAL BLOCKWISE HESSIAN EVALUATION

Neural networks typically have a large number of parameters, requiring approximations to the Hessian due to the massive computational overhead for producing the actual values. Diagonal approximations are employed to capture curvature information relevant for optimization (Becker et al., 1988, Elsayed & Mahmood, 2022, LeCun et al., 1989), but this approximation is too coarse-grained and over-estimates the rank. There are low-rank approximations of the Hessian (Roux et al., 2007), these too are problematic because we aim to measure the rank of the Hessian. The empirical Fisher information approximates the gradient using the outer-product of gradients, but has been shown to not capture curvature information, especially away from local minima (Kunstner et al., 2019). Other approximations that use the Fisher information matrix require stochastic models, which limit their applicability (Martens & Grosse, 2015)

Our approach, a partial blockwise Hessian, builds off recent work studying the layerwise Hessian (Sankar et al., 2021). Even if we could calculate and store an approximation to the full Hessian, we would not be able to calculate the srank because a singular value decomposition has a cubic-time complexity. At the same time, a purely layerwise approximation of the Hessian cannot capture changes in the Hessian due to the target changing for piece-wise linear activations like `ReLU` and `leaky-ReLU`. We denote the layer-specific parameters by $\theta^{(l)}$ and the entire parameter set as $\theta$, then the Hessian with respect to $\theta^{(l)}$ for piece-wise linear activations is independent of $y$, $\frac{d}{dy}\nabla_{\theta^{(l)}}^2 J(\theta) = 0$, making the layerwise Hessian an unsuitable for our empirical analysis.

The partial blockwise Hessian is the exact Hessian for a subset of the layers. We argue that the blockwise Hessian is a reasonable choice because the layerwise Hessian was shown to approximate the statistics of the entire Hessian (Wu et al., 2020). Taking the Hessian with respect to parameters of more than one layer allows us to to analyze the Hessian at the task change boundary, because $\frac{d}{dy}\nabla_{\theta^{(l)},\theta^{(l-1)}}^2 J(\theta) \neq 0$. In practice, we take the blockwise Hessian with respect to the parameters of the last 2 layers because we found it sufficient to capture curvature changes while being small enough to calculate the singular value decomposition throughout training.

## 5 Preserving Curvature Throughout Continual Learning

In the previous section, we claimed that loss of curvature can explain loss of plasticity. If curvature is lost over the course of learning, then one solution to this problem could be to regularize towards the curvature present at initialization. While explicit Hessian regularization would be computationally costly, previous work has found that even L2 regularization can mitigate loss of plasticity (Dohare et al., 2021, Kumar et al., 2023, Lyle et al., 2021), without attributing this benefit to preserving curvature. These methods, however, do more than just prevent loss of curvature, they also prevent parameters from growing large (subject to the regularization parameter's strength). L2 regularization and weight decay, for example, mitigate plasticity loss but also prevent the parameters from deviating far from the origin. This could limit the types of functions that the neural network can learn, requiring the regularization strength to be tuned.

We propose a new regularizer that is simple and that gives the parameters more leeway for moving from the initialization, while preserving the desirable plasticity and curvature properties of the initialization. Our regularizer penalizes the distribution of parameters if it is far from the distribution of the randomly initialized parameters. At initialization, the parameters at layer $l$ are sampled i.i.d. $\theta_{i,j} \sim p^{(l,0)}(\theta)$ according to some pre-determined distribution, such as the Glorot initialization (Glorot & Bengio, 2010). During training, the distribution of parameters at any particular layer is no longer known and the parameters are neither independent nor identically distributed. However, it is still possible to regularize the empirical distribution towards the initialization distribution by using the empirical Wasserstein metric (Bobkov & Ledoux, 2019). We denote the flattened parameter matrix for layer $l$ at time $t$ by $\bar{\theta}^{(l,t)}$, then the squared Wasserstein-2 distance between the distribution of parameters at initialization and the current parameter distribution is defined as,

$$\mathcal{W}_2^2\left(p^{(l,0)}, p^{(l,t)}\right) = \frac{1}{d}\sum_{i=1}^{d}\left(\bar{\theta}_{(i)}^{(l,t)} - \bar{\theta}_{(i)}^{(l,0)}\right)^2.$$

The order statistics of the paramater is denoted by $\theta_{(i)}^{(l,t)}$ and represents the $i$-th smallest parameter, which is sorted independently from the initialization. In the above equation, we are taking the L2 difference between the order statistics of each layer's parameters at initialization and a point in training. The *Wasserstein initialization regularizer* uses the empirical Wasserstein distance for each layer of the neural network.

A recent alternative, regenerative regularization, regularizes the neural network parameters towards their initialization (Kumar et al., 2023). This regularizer mitigates plasticity loss, but it also prevents the neural network parameters from deviating far from the initialization. The difference between the Wasserstein initialization regularization and the regenerative regularizer is the fact that we take the difference in the order statistics. As we will show, however, the Wasserstein regularizer also mitigates plasticity loss but allows the neural networks parameters to deviate far from the initialization.

## 6 Experiments: Curvature Changes in Plasticity Benchmarks

We now investigate our claim that loss of curvature, as measured by the reduction in the rank of the Hessian, explains loss of plasticity. Our experiments use the three most common continual learning benchmarks that exhibit loss of plasticity:

- MNIST with periodic pixel observation permutation, a commonly used benchmark across continual learning (Dohare et al., 2023a, Elsayed & Mahmood, 2023, Goodfellow et al., Kumar et al., 2023, Zenke et al., 2017).

- MNIST with periodic label shuffling, an increasingly used variant of permuted MNIST and noted for its increased difficulty (Elsayed & Mahmood, 2023, Kumar et al., 2023, Lyle et al., 2023). This is the problem setting studied in Section 3.

- CIFAR-10 (Krizhevsky, 2009) with periodic label shuffling. This is an increasingly common problem setting for plasticity, and it is difficult due to the relative complexity of image observations in CIFAR and the difficulty of relearning labels (Kumar et al., 2023, Lyle et al., 2023, Sokar et al., 2023).

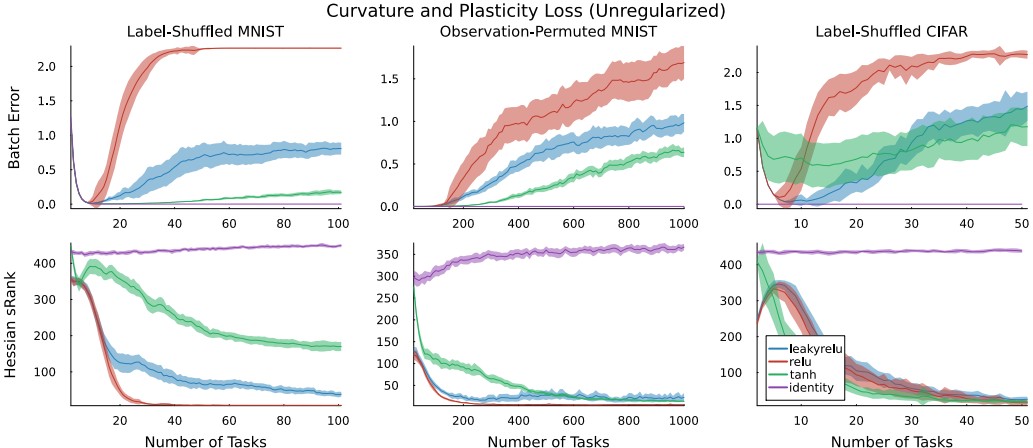

Figure 2: Results with unregularized objectives. Top Row: Batch Error at task end. Bottom Row: sRank of Hessian at task-beginning. Left: Label-shuffled MNIST. Middle: Observation-permuted MNIST. Right: Label-shuffled CIFAR-10.

To provide evidence of the claim that curvature explains loss of plasticity, we conduct an in-depth analysis of the change of curvature in continual supervised learning. We first show that curvature is a consistent explanation across different problem settings. Afterwards, we show that the gradient stays contained in the top-subspace of the Hessian, which shrinks over the course of continual learning. Lastly, we show that regularization which has been demonstrated to be effective in mitigating loss of plasticity also mitigates loss of curvature.

## 6.1 DOES LOSS OF CURVATURE EXPLAIN LOSS OF PLASTICITY?

We present the results on the three problem settings in Figure 2. Like the results in Section 3, loss of plasticity occurs in problem settings when non-linear activations are used. Loss of curvature tends to co-occur with loss of plasticity for the non-linear activations, providing a consistent explanation of the phenomenon compared to previous explanations. There is also some evidence that loss of curvature may precede loss of plasticity. In the label-shuffled MNIST experiment, `tanh` loses curvature before the error begins to increase. This finding suggests that loss of curvature may be the underlying cause of plasticity loss.

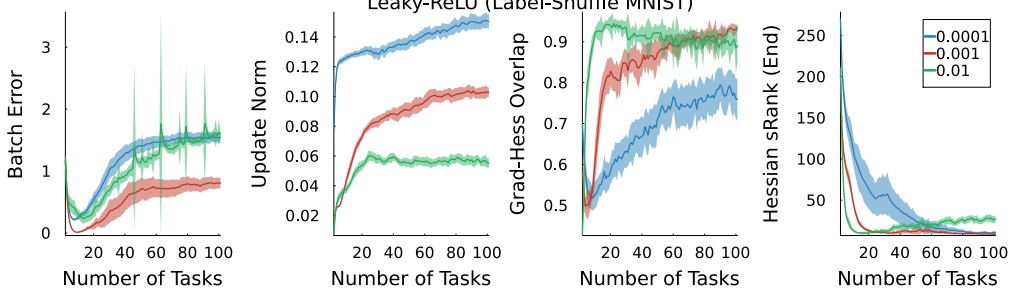

Figure 3: Label-shuffled MNIST with `leaky-ReLU` and different step-sizes. From left to right: task-end batch error, average update norm, gradient-hessian overlap at task-end and task-end Hessian rank.

## 6.2 HOW DOES LOSS OF CURVATURE AFFECT LEARNING?

Having demonstrated that loss of curvature co-occurs with loss of plasticity, we now investigate how loss of curvature affects the gradients and learning. Our goal is to explain why the update norms can be increasing despite loss of plasticity. We focus on `leaky-ReLU` with different step-sizes on

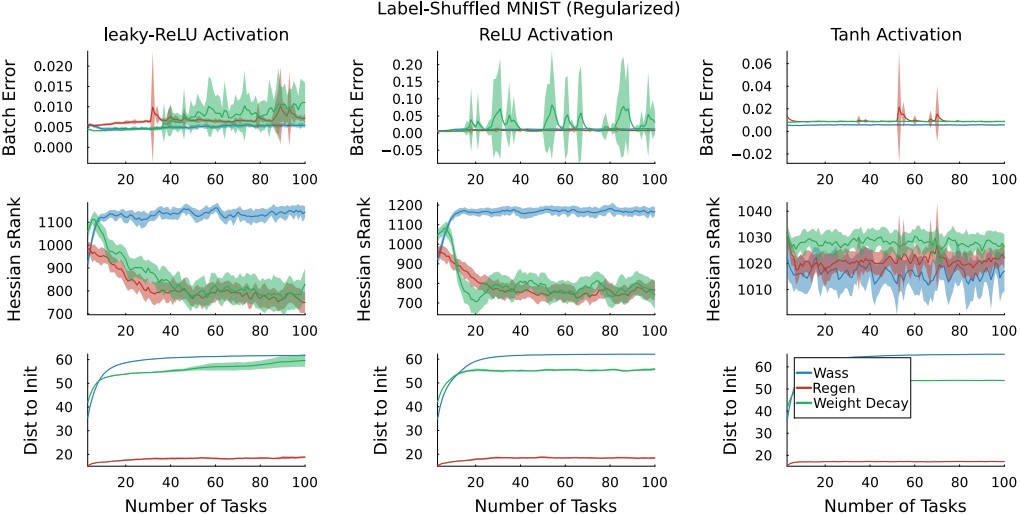

Figure 4: Results on label-shuffled MNIST with regularization. All regularizers mitigate loss of plasticity and loss of curvature to some degree but weight decay is unstable. Top: Batch training error at the end of task. Middle: Hessian srank at beginning of task. Bottom: Distance from current parameters to original random initialization.

the label-shuffled problem, which exhibits loss of plasticity at every step-size but an increase in the average update norm over training. In Figure 3, we plot the overlap of the gradient and the Hessian-gradient product at task-end, given by $\frac{g^T H g}{\|g\|\|Hg\|}$, which measures whether the gradient is contained in the top subspace of the Hessian (Gur-Ari et al., 2018). Gradients at the end of the task tend to be contained in the top-subspace of the Hessian which is also decreasing in rank. We hypothesize that the update norm is increasing because the gradients along the small subspace are more likely to add up to a large momentum, but that this is not enough to escape the flat initialization on the subsequent tasks. This is evidenced by the stabilization of the update-norm for step-size $\alpha = 0.01$ which co-occurs with the increase in the task-end Hessian rank. See Appendix C.2 for more examples of this phenomenon.

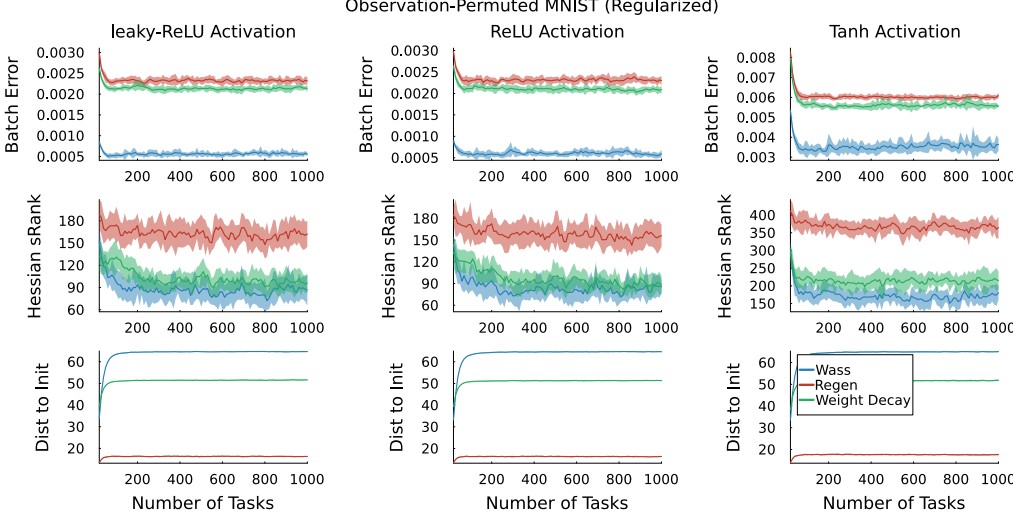

Figure 5: Results on observation-permuted MNIST with regularization. The Wasserstein regularizer achieves a lower error, is able to travel further from its initialization and maintains curvature.

### 6.3 CAN REGULARIZATION PRESERVE CURVATURE?

We now investigate whether regularization prevents loss of plasticity and, if it does, whether it also prevents loss of curvature. Our results for the three problem settings and activation functions are summarized in Figures 4, 5, and 6. We see that all regularizers are able to prevent plasticity loss to some degree. L2 regularization is the least performant, and it is often unstable. Across activation functions, we find that the Wasserstein regularizer is able to reliably achieve a lower error compared to the other regularizers. The benefit of the Wasserstein regularizer can be seen from the bottom row of plots: the parameters are able to deviate more from the initialization compared to the other regularizers. Additionally, the Wasserstein regularizer is less sensitive to the hyperparameter controlling the regularization strength (see Appendix C.5).

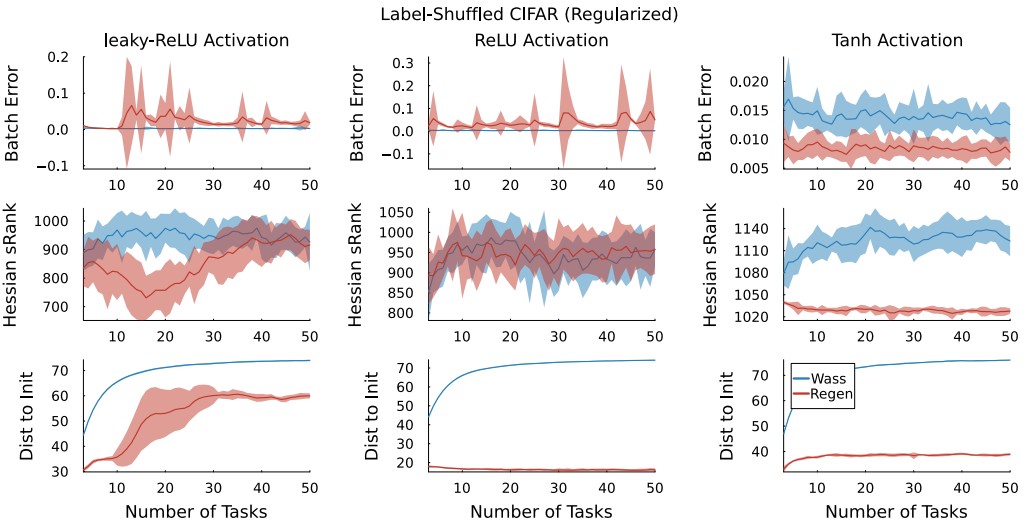

Figure 6: Results on label-shuffled CIFAR with regularization. Regenerative regularization is unstable with piecewise linear activations, but is comparable to the Wasserstein regularizer for `tanh` activations. Due to instability, results with weight decay can be found in Appendix C.3.

## 7 DISCUSSION

We have demonstrated how loss of curvature is a more consistent explanation for loss of plasticity when compared to previous explanations offered in the literature. One limitation of our work is that we study an approximation to the Hessian. Our experiments suggest that this approximation of the Hessian is enough to capture changes in curvature, but more insight may be found from theoretical study of the entire Hessian. Another limitation is that it is not clear what drives neural networks to lose curvature during training. Understanding the dynamics of training neural networks with gradient descent, however, is an active research area even in supervised learning. It will be increasingly pertinent to understand what drives neural network training dynamics to lose curvature so as to develop principled algorithms for continual learning.

Our experimental evidence demonstrates that, when loss of plasticity occurs, there is a reduction in curvature as measured by the rank of the Hessian at the beginning of subsequent tasks. When loss of plasticity does not occur, curvature remains relatively constant. Unlike previous explanations, this phenomenon is consistent across different datasets, non-stationarities, step-sizes, and activation functions. Lastly, we investigated the effect of regularization on plasticity, finding that regularization tends to preserve curvature but can be sensitive to the regularization strength. We proposed a simple distributional regularizer that proves effective in maintaining plasticity across the problem settings we consider, while maintaining curvature and being less hyperparameter sensitive.

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

APPENDIX

## A ADDITIONAL ANALYSIS ON COUNTER-EXAMPLES

In the body of paper, we provided a high-level partial analysis on Figure 1, and concluded that each metric corresponding to previous explanations implies the inconsistency with the loss of plasticity (i.e. increase in training error). Here, we aim to compliment that high-level analysis on Figure 1 by providing detailed explanation on how each metrics are inconsistent with the batch error.

1. **Average Update Norm** (top-middle): Plot measures the average L1 norm of the parameter updates for each task. `Leaky-ReLU` and `ReLU` shows the completely opposite trend. The former constantly increases from 0.03 up to 0.1, while the latter decreases from 0.025 to 0. Other activations, `identity` and `tanh`, remains at the level of 0.03. Since `leaky-ReLU` suffers from the plasticity loss, the fact that `leaky-ReLU` experiencing the increase in update norm is inconsistent.

2. **Stable Rank of Representation** (top-right): The plot measures the stable rank of the representation for each task. Among all the activations, `identity` retains the highest level of rank throughout the experiment. It remains around the level of 30. Activations except for `identity` initially holds an stable rank of 25. While `tanh` sustain its rank over the tasks, `ReLU`/`leaky-ReLU` face with the distinct levels of drop. `ReLU` drastically drops its representation rank to 5. On the other hand, the rank of `leaky-ReLU` eventually converge to 20. The representation rank is inconsistent because `tanh` retains its representation rank at 25, while it still experience plasticity loss.

3. **Dormant Nuerons** (bottom-left): Neuron dormancy are measured by the entropy of the absolute value of the features for each task, which captures the notion of dormancy that activations can concentrate on a small subset of features. The plot shows that the activations other than `ReLU` retains its level of activation entropy. Precisely, `identity` and `tanh` are overlapping at the level of 3.5, while `leaky-ReLU` remains at the level of $2.6 - 2.7$. Both `tanh` and `leaky-ReLU` experience plasticity loss but the neuron dormancy is non-decreasing, making the explanation inconsistent.

4. **Weight Norm** (bottom-middle): The plot presents the L1 norm of the weights at the end of each task. For all the activations, the weight norm is $0.05 - 0.1$ at the beginning, and monotonically increase as task progresses. In order for weight norm to be a consistent metric, it must correlate to the trend of batch error. However, our result disagree with this criteria, as the weight norm for `ReLU` (which has the worst batch error) only hit the 0.1 at the last task while others reach to the higher value.

5. **Stable Rank of Weights** (bottom-right): Similarly to the stable rank of representation, the plot tracks change in the stable rank of model weights per task. With `leaky-ReLU`, there is a drop in the weight rank from 32 to 31, but all other activations concentrate and remain at the level of 32. This suggests that the weight rank has very little to do with plasticity.

## A.1 CORRELATION PLOTS

In this section, we present the same results as in Section 3 in the form of a correlation plot. In the correlation plot, we plot the batch error at the end of a task on the x-axis and a measurement that explains plasticity on the y-axis. We then plot the line of best fit for those points for each activation function. This provides a quantiative analysis of the previous explanations of plasticity loss. We also include the correlation plot with the sRank of the Hessian, which was presented in Section 4.

We are interested in a consistent trend, where the slope of the fitted line does not change for the different activation functions. In Figure 7, we see that the only explanation that exhibits a consistent trend is the Hessian sRank. The previous explanations of plasticity are inconsistent because they exhibit mixed positive, zero and negative correlations.

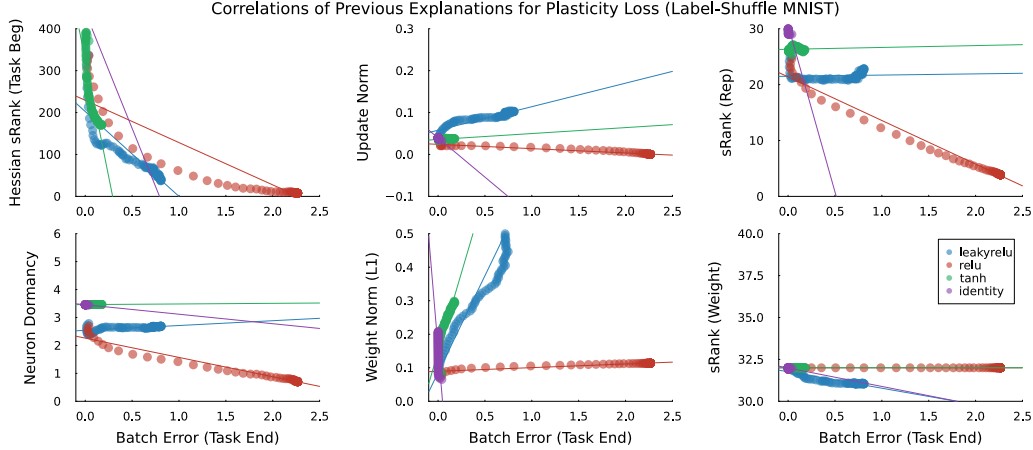

Figure 7: Correlation plots for different explanations of plasticity loss. The points plot pairs of (batch error, plasticity measurement). The line is fit with a simple linear regression to the points. The color of the line and plot correspond to the activation function. The only consistent explanation is the Hessian sRank, demonstrating that higher rank Hessian's at the beginning of a task lead to lower errors at the end of the task. Previously proposed explanations exhibit mixed positive, zero and negative correlations.

# B    EXPERIMENTAL DETAILS

## B.1    LABEL-SHUFFLED MNIST

Non-stationary variant of the ordinary (stationary) supervised classification problem on MNIST dataset. The source of non-stationarity in this problem is the periodical random shuffling of labels, irrespective of the original class labels. The dataset consists of 1280 uniformly sampled MNIST image-label pairs. We iterate over the dataset for 200 epochs in the experiments in the main paper, but ablate for different number of epochs in C.1. After 200 epochs, the labels will be reshuffled within the same dataset, producing the new task. Each gradient updates are performed with the batch of 16 datapoints, hence the update capacity per task is 80. The total number of tasks is 100. The architecture is a 3 hidden layer feed-forward neural network with widths $(128, 64, 32)$. We use the Adam optimizer with default hyperparameters, and sweep over the learning rates $\{0.01, 0.001, 0.0001\}$ We average over 30 seeds for the unregularized experiments and average over 10 seeds for the regularized experiments. For the regularized experiments, we sweep over the regularization strength of $\{1.0, 0.01, 0.001, 0.0001\}$.

## B.2    OBSERVATION-PERMUTED MNIST

The overall problem framework is identical to the label-shuffled MNIST, except for the source of non-stationarity. The non-stationarity is introduced by reordering the positions of pixels in each input image, while label remains the same throughout the experiment. At the beginning of each task, the permutation of pixels are shuffled, and each input images are uniformly shuffled according to that permutation. Other basic components of experiment does not vary from label-shuffled MNIST problem.

## B.3    LABEL-SHUFFLED CIFAR

A non-stationary supervised classification problem using the CIFAR dataset, similar to the label-shuffled MNIST problem. The only difference from label-shuffled MNIST problem are the use of CIFAR10 dataset and the number of tasks to evaluate on. Similarly in the label-shuffled MNIST problem, this problem uniformly samples 1280 datapoints from CIFAR10. The batch size remainns the same, and so update budget does. Number of tasks, however, is reduced to 50 due to the increase in computational cost and because the problem difficulty is higher, resulting in plasticity loss sooner. The architecture uses 3 convolutional layers with stride 2 and $(16, 32, 64)$ filers respectively before flattening and using a 2 layer feed-forward neural network with widths $(64, 32)$. We use the Adam optimizer with default hyperparameters, and sweep over the learning rates $\{0.01, 0.001, 0.0001\}$ We average over 30 seeds for the unregularized experiments and average over 10 seeds for the regularized experiments. For the regularized experiments, we sweep over the regularization strength of $\{1.0, 0.01, 0.001, 0.0001\}$.

## B.4    CONTINUAL IMAGENET

We use the Continual ImageNet environment proposed by Dohare et al. (2023a). While we train the same convolutional neural network described by Dohare et al. (2023a), we use a batch size of 100 and train for 200 epochs with Adam using a step size of 0.001. The regularization strength used was 0.01, dropout rate is 0.5 and we reset the last layer at the beginning of each new task. Our results are averaged over 20 seeds.

# C    ADDITIONAL RESULTS

## C.1    UPDATE BUDGET EFFECT ON PLASTICITY

By varying the number of epochs in a task, the neural network is able to learn more on a task, perhaps allowing the neural network to escape loss of plasticity. Unfortunately, increasing the number of epochs only marginally delays the onset of loss of plasticity. Plasticity loss still occurs, but reduction in curvature is a consistent predictor of this phenomenon.

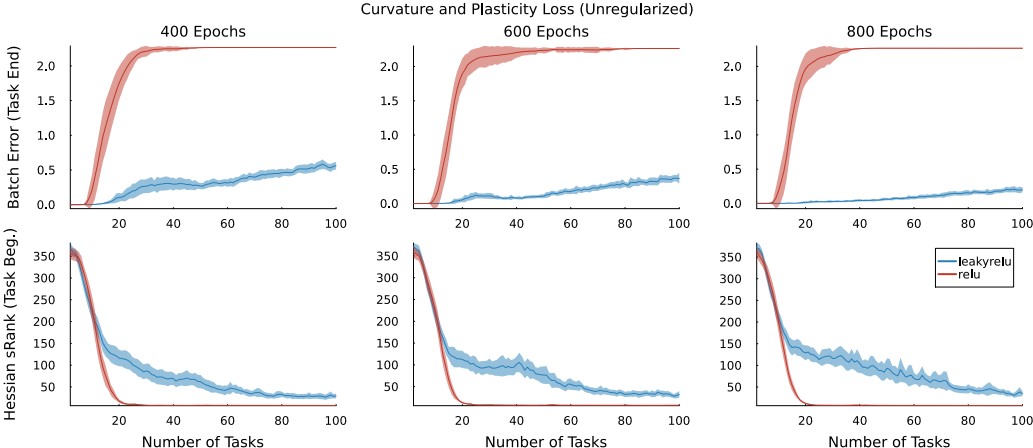

Figure 8: Ablating number of epochs per task on label-shuffled MNIST

## C.2 LOSS OF CURVATURE'S EFFECT ON UPDATES

We provide more results on the update norm, Hessian rank, and Hessian-Gradient overlap at the beginning and end of a task, as well as across more problem settings. The findings are consistent that the update norm is influenced by whether the gradient overlaps with the top-subspace of the Hessian and the size of that subspace.

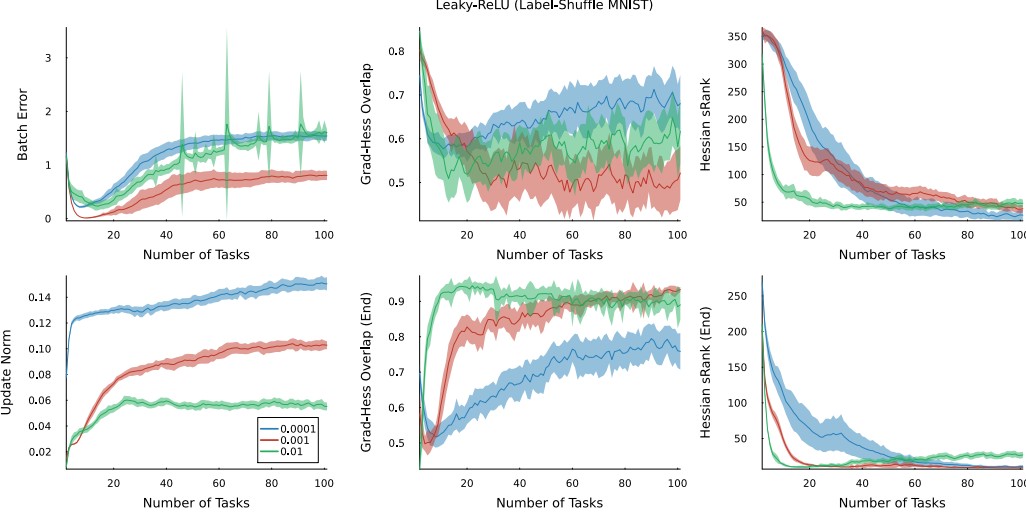

Figure 9: Explaining why neural networks with `leaky-ReLU` lose plasticity despite an increasing update norm in label-shuffled MNIST.

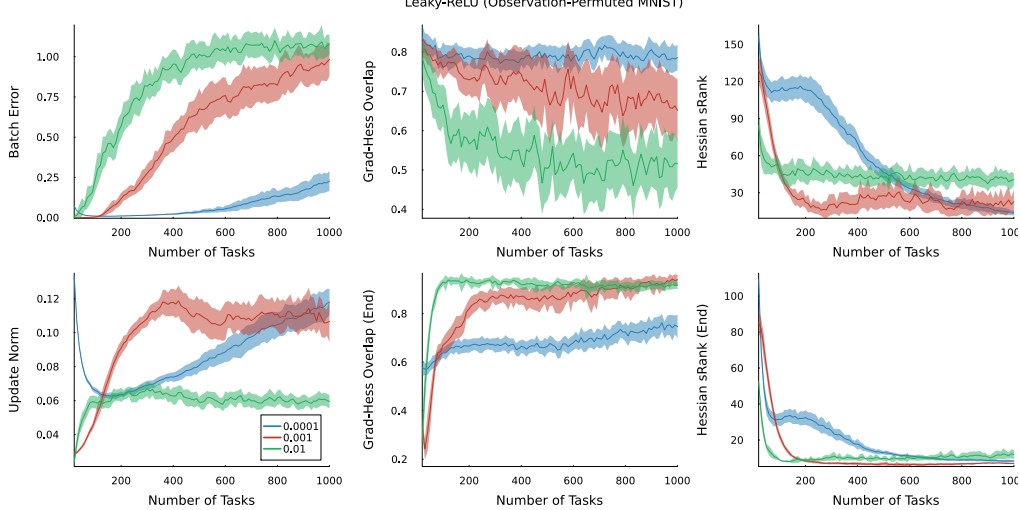

Figure 10: Explaining why neural networks with `leaky-ReLU` lose plasticity despite an increasing update norm in observation-permuted MNIST.

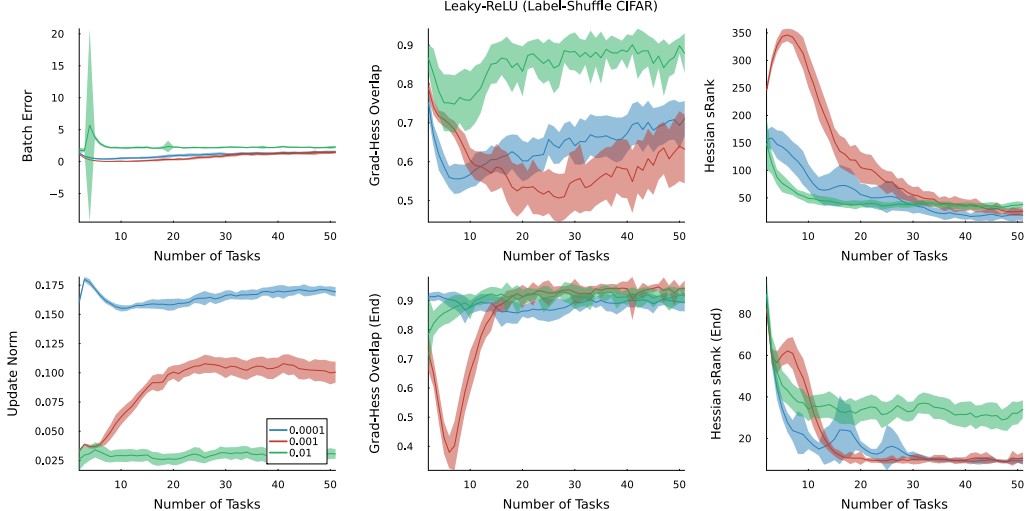

Figure 11: Explaining why neural networks with `leaky-ReLU` lose plasticity despite an increasing update norm in label-shuffled CIFAR.

### C.3 CIFAR RESULTS INCLUDING WEIGHT DECAY

We omitted the weight decay results for CIFAR in the main paper due to instability. Notably, weight decay exhibits the biggest reduction in its curvature with `leaky-ReLU` where it is most unstable.

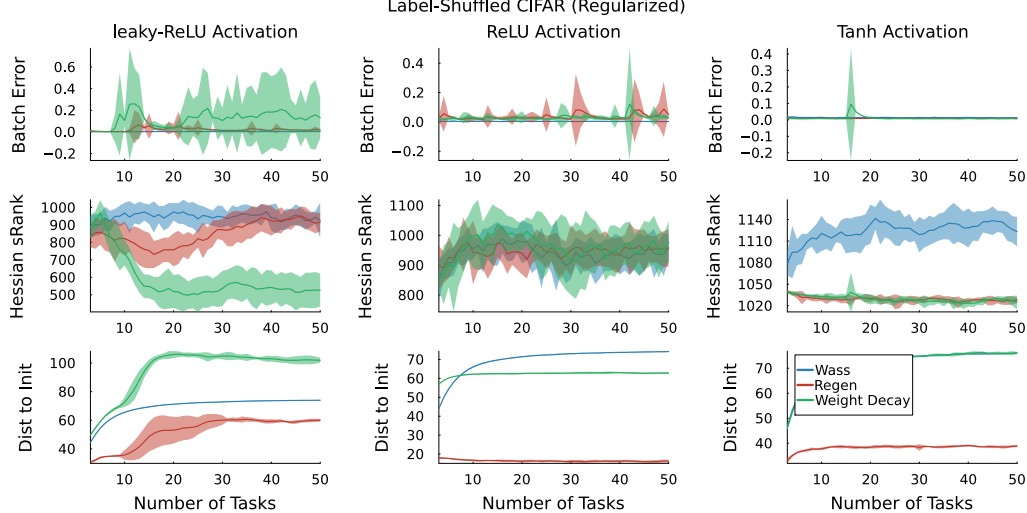

Figure 12: Results on label-shuffled CIFAR with regularization. Regenerative regularization is unstable with piecewise linear activations, but is comparable to the Wasserstein regularizer for `tanh` activations.

## C.4 ONLINE ERROR CAN BE INFLUENCED BY HIGH START ERROR

Average online error is another metric for studying loss of plasticity, but it can misdiagnose the phenomenon. Even if a neural network maintains a consistent error at the end of a task, its average online error can increase due to an increase in its error at the beginning of a task. But the error at the beginning of a task is not controllable, because it is due to a non-stationarity in the experience. Thus, we focus on the batch error at task end alone.

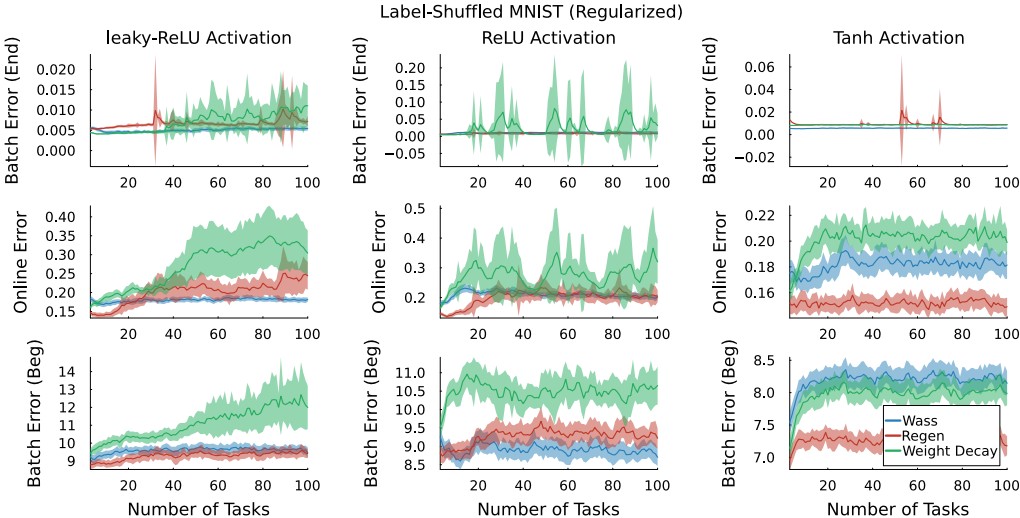

Figure 13: Average online error can misdiagnose loss of plasticity due to an increase in the start error.

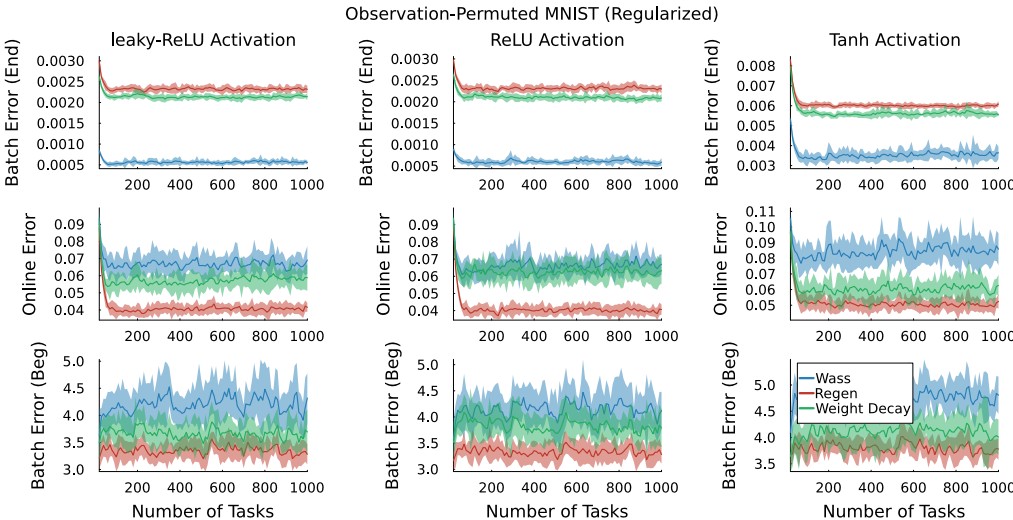

Figure 14: Even if the start-error is constant, average online error has an implicit bias towards algorithms that start with low error, but this is not something the algorithm itself can control or optimize for because it is due to a task change.

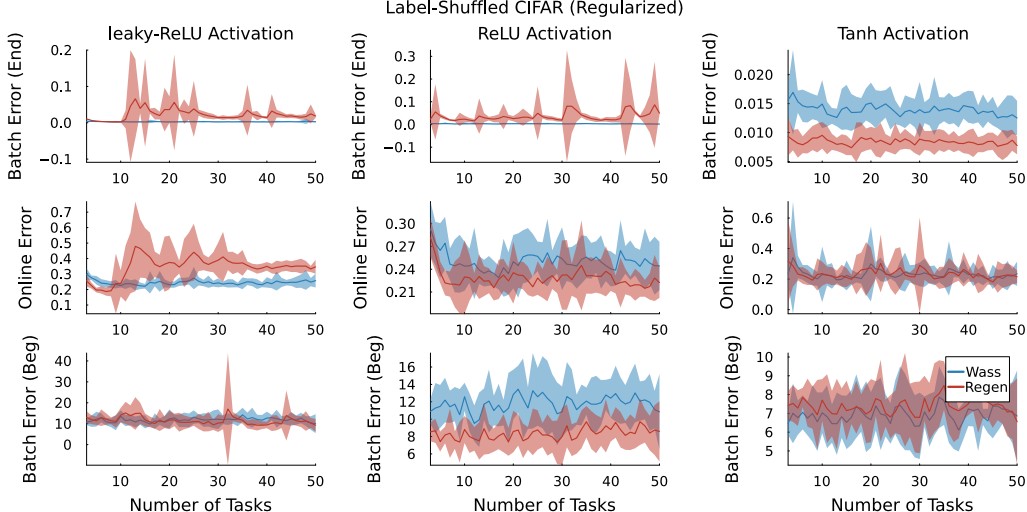

Figure 15: Average online error can fail to distinguish between algorithms who achieve different performance levels because of high error at the start of a task.

## C.5   REGULARIZER HYPERPARAMETER SENSITIVITY

The plots below show the batch error at the end of a task for different regularization strengths. Compared to weight decayy and regenerative regularization, the Wasserstein regularizer is able to reach and maintain a lower error across most problems and activation functions.

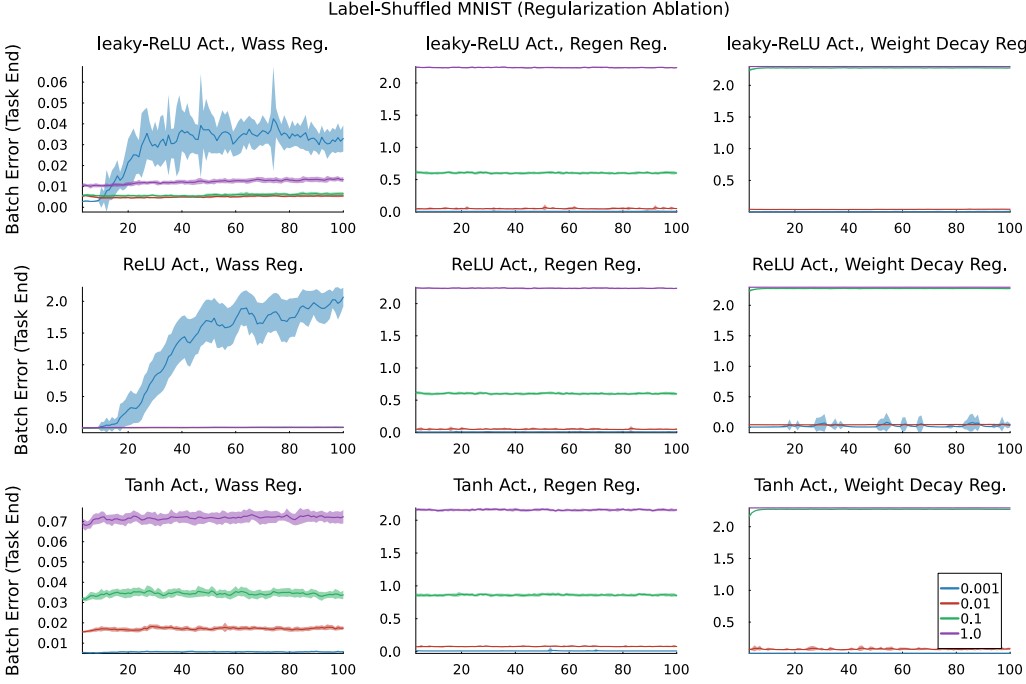

Figure 16: Hyperparameter ablation for label-shuffled MNIST.

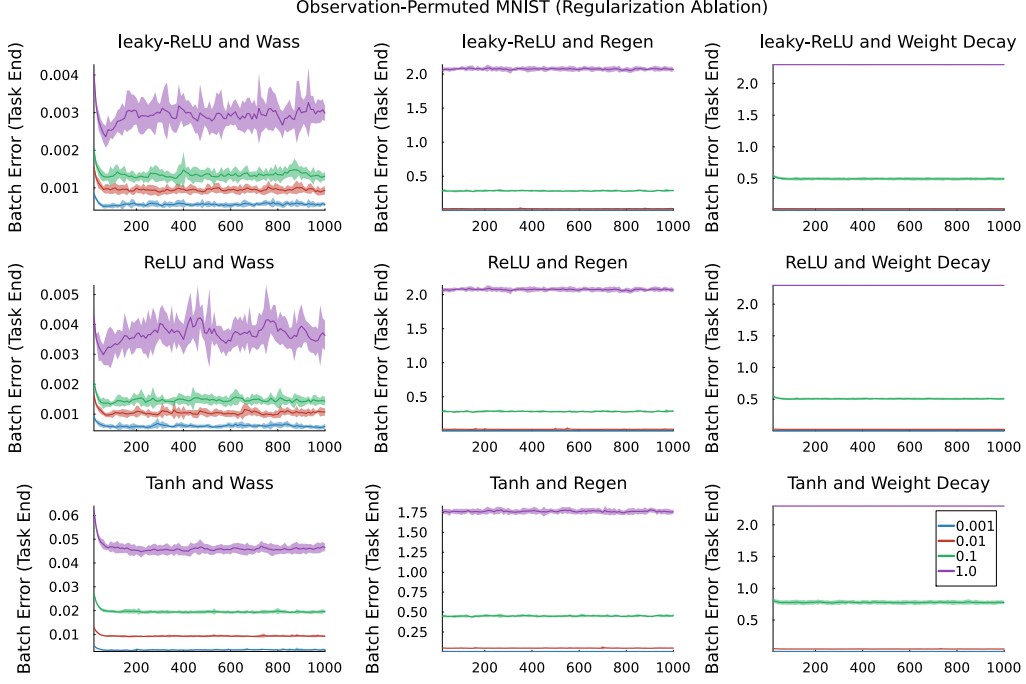

Figure 17: Hyperparameter ablation for observation-permuted MNIST.

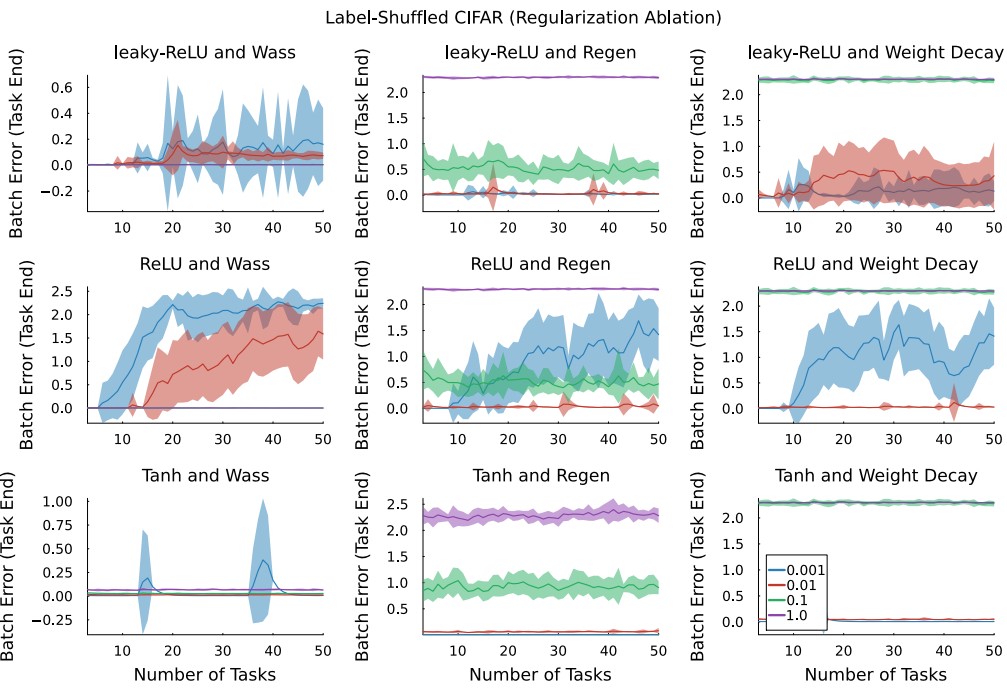

Figure 18: Hyperparameter ablation for label-shuffled CIFAR.

# D    RESULTS ON CONTINUAL IMAGENET

We conduct a thorough experimental study on Continual Imagenet. Our analysis in the first subsection follows the analysis in the main paper. The next subsection studies the effect of neural network modifications. The next two subsections analyze the correlations of the Hessian sRank with plasticity loss, showing a consistent trend, while demonstarting that the sharpness of the Hessian does not show a consistent trend.

## D.1    PLASTICITY AND CURVATURE WITH REGULARIZERS

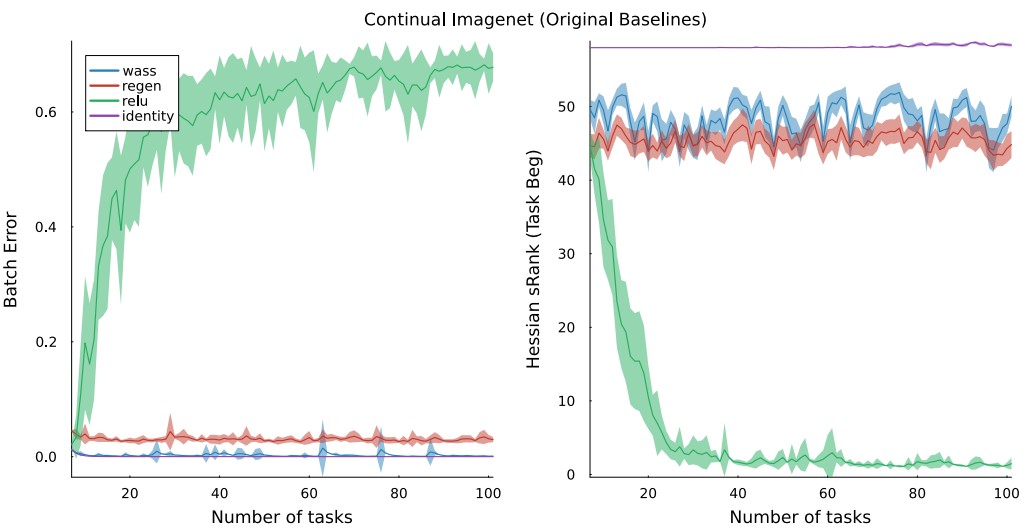

Figure 19:   Both the Wasserstein and Regenerative regularizer are able to mitigate loss of plasticity exhibited by deep networks with relu activations and preserve curvature. We again find that the Wasserstein regularizer is able to achieve lower error than regenerative regularization.

## D.2 Plasticity and Curvature with Neural Network Modifications

We compare the Wasserstein regularizer against several neural network modifications that have been shown to help reduce loss of plasticity. In particular, we use `CReLU` (Abbas et al., 2023, Shang et al., 2016), layernorm (Ba et al., 2016, Lyle et al., 2023), dropout (Srivastava et al., 2014) and last layer reset (Nikishin et al., 2022)

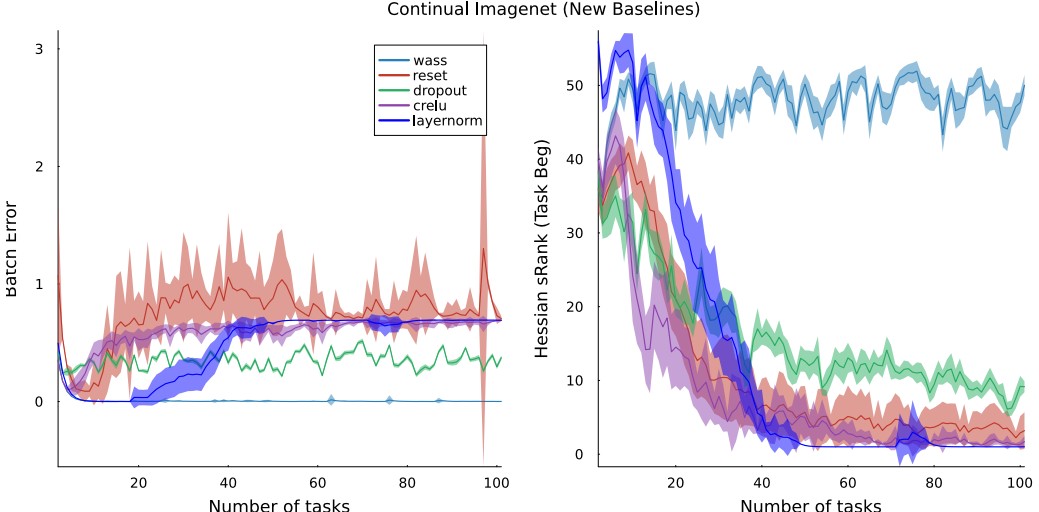

Figure 20: Comparing against neural network modifications, and including the Wasserstein regularizer. Dropout is not able to learn to solve any task. `CReLU`, last layer reset and layernorm fail to mitigate loss of plasticity in this problem. All of these network modifications also fail to preserve curvature.

### D.3 ADDITIONAL CORRELATION PLOTS WITH SRANK

We provide correlation plots for the batch error at the end of the task and the sRank of the Hessian. The color of the line and plot correspond to different activation functions, regularizers and/or neural network modifications. The plotted points are pairs of (batch error, srank of Hessian). The line is fit with a simple linear regression.

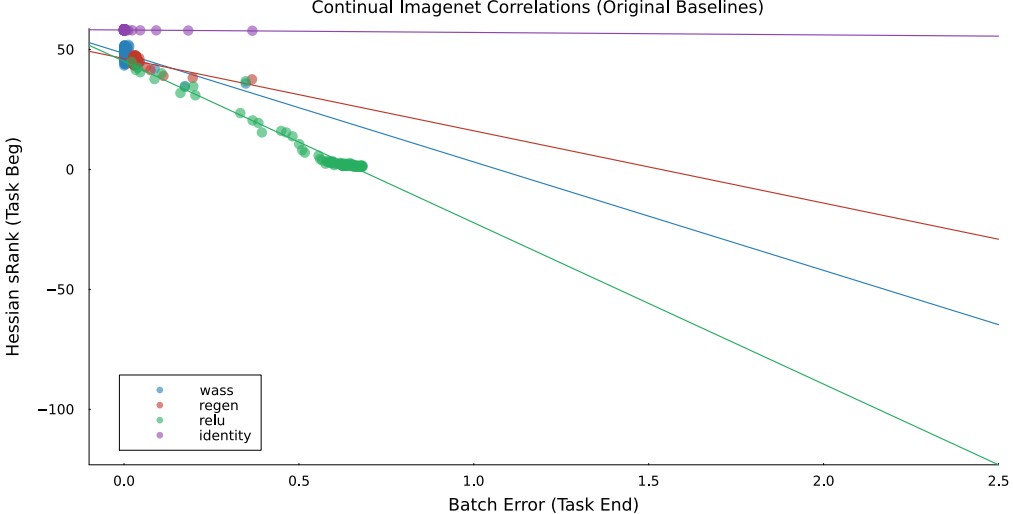

Figure 21: The Hessian sRank explains loss of plasticity for all non-linear activation functions while also demonstrating that the linear function does not exhibit loss of plasticity.

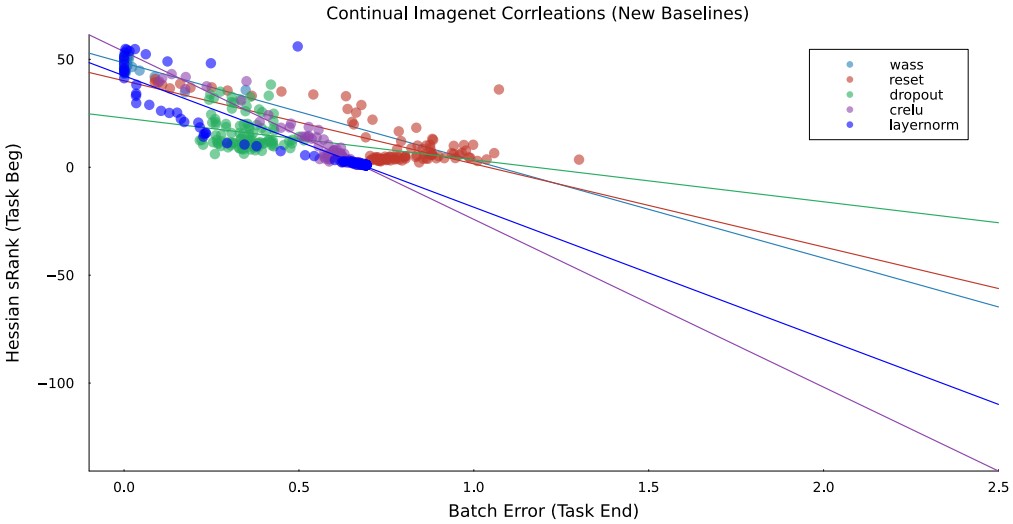

Figure 22: The Hessian sRank also consistently explains loss of plasticity for the neural network modifications.

We provide correlation plots for the batch error at the end of the task and the sharpness (the maximum singular value of the Hessian). The color of the line and plot correspond to different activation functions, regularizers and/or neural network modifications. The plotted points are pairs of (batch error, maximum singular value of Hessian). The line is fit with a simple linear regression.

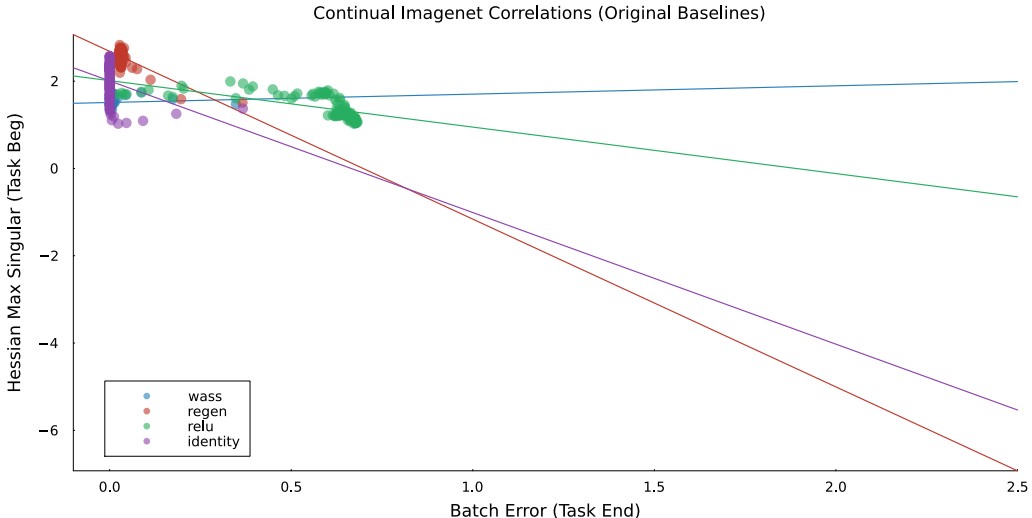

Figure 23:   For the regularizers, sharpness shows a similar trend to the rank. One issue is that the line of best fit for identity has a slight positive slope.

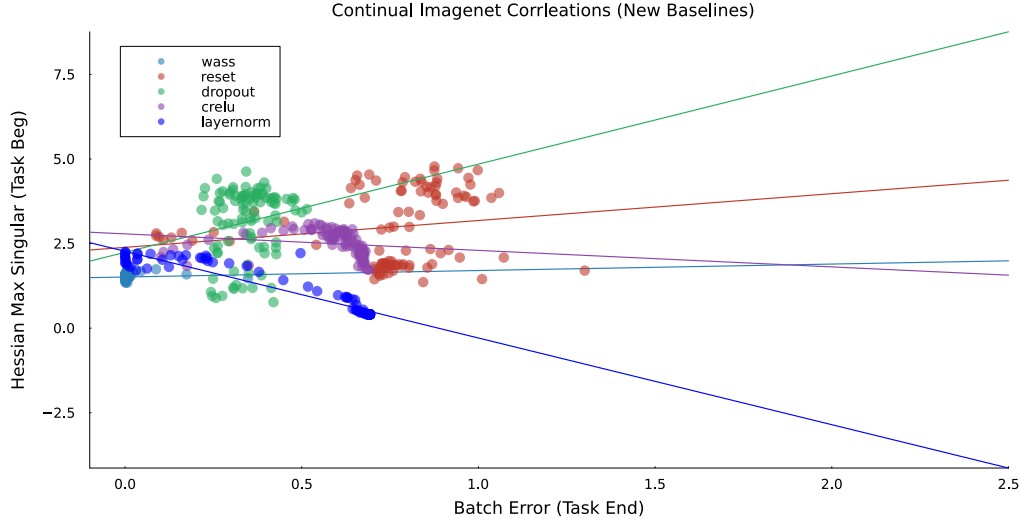

Figure 24:   Sharpness is not a good explanation for loss of plasticity as it exhibits mixed positive, zero and negative correlations.

## D.5 INTER-TASK ONLINE LEARNING CURVES

Laslty, we look at the early learning behavior within the first 10 tasks for the regularizers. We find that the regularizers do not slow learning down, and that they do benefit from non-linear feature representations and achieve, locally, a lower error than the linear function. Note that the online error plotted here is with respect to a changing model, and thus the results cannot be compared to the batch error at the end of the task in a straighforward way.

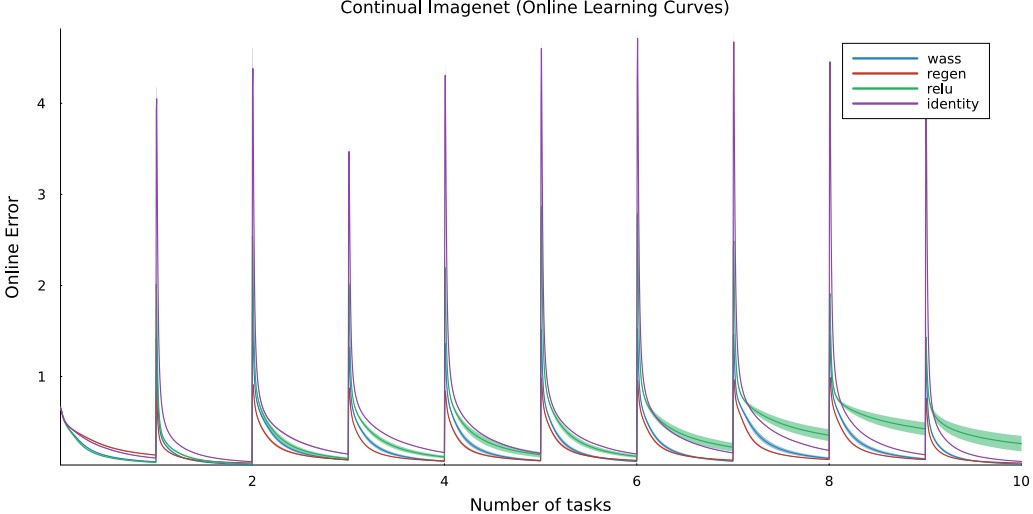

Figure 25: Online error over the first 10 tasks. ReLU and Wasserstein regularized ReLU all attain a lower error on the first task compared to the deep linear network. As the number of tasks increases, ReLU loses plasticity but both regularizers are able to maintain plasticity and benefit from non-linear representations.

# E    RESULTS ON MNIST WITH LARGER SUBSET

We expand the results on MNIST to a larger subset of images. Although loss of plasticity occurs on a subset of 1280 MNIST images, this larger MNIST subset of 12800 is more difficult to fit: a linear model with 784*10 (input dim × output dim) effective parameters can no longer achieve a low error. The results in this experiment are similar but more pronounced than in the smaller subset of MNIST. The linear model does not lose plasticity but it also can no longer achieve low error. The non-linear activation functions can achieve a much lower error in the first task but, due to loss of plasticity, eventually becomes worse than the linear model. Regularization mitigates loss of plasticity, and loss of plasiticity is well explained by the Hessian sRank. Feature rank and Hessian sRank exhibit no clear relationship.

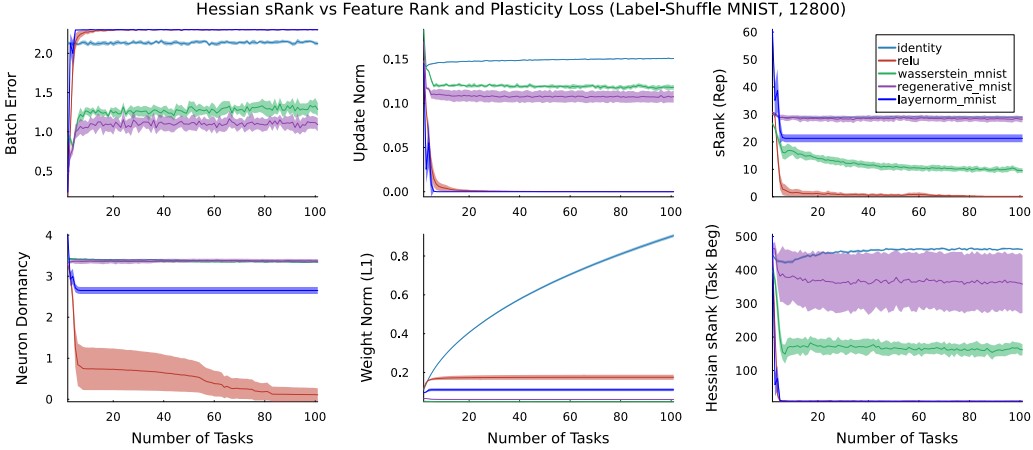

Figure 26:   Larger subsets of MNIST are more difficult to fit using a linear model. Curvature, as measured by the Hessian sRank, remains the best explanation of loss of plasticity. Regularization also remains a good mitigator for loss of plasticity. Like the smaller subset, regenerative regularizer does marignally better on this problem and requires more extensive hyperparameter tuning.

## F    RESULTS ON MNIST WITH 13 LAYER NEURAL NETWORK

We validate our blockwise approximation to the Hessian sRank by using a much deeper 13 layer neural network and demonstrating the same consistent trend. The architecture is similar to the previous MNIST experiment described in Appendix B.1, except we have 10 additional layers with 128 neurons. Our results in Figure 27 shows that our blockwise aproximation to the Hessian sRank exhibits the same downward trend when loss of plasticity occurs and constant trend when linear or regularized non-linear.

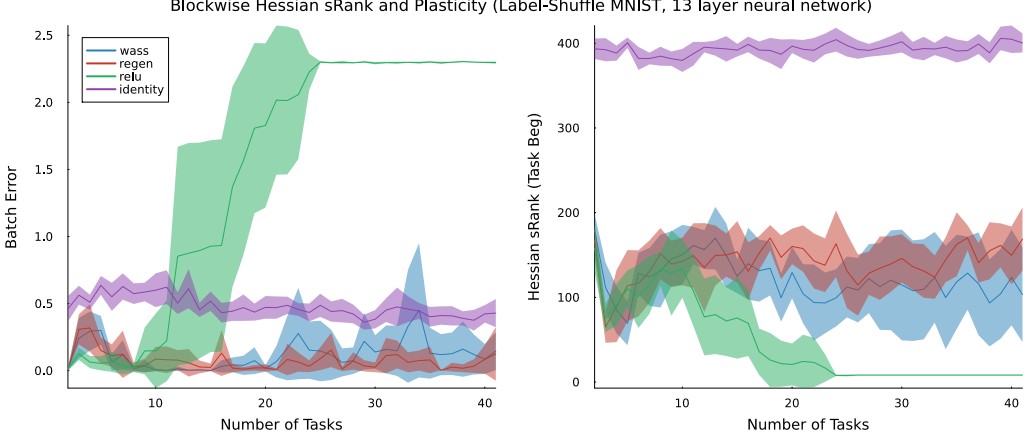

Figure 27:   A 13 layer neural network exhibits the same trend in our blockwise approximation to the Hessian. The larger capacity relu network still suffers from loss of plasticity, and this is reflected in the reduction of the Hessian sRank. The linear network maintains a moderate error and both regularized networks can achieve lower error, all while maintaining the Hessian sRank.

