# OpenReview forum: "Curvature Explains Loss of Plasticity"
_ICLR.cc/2024/Conference — Submitted to ICLR 2024_

### Official Review · Reviewer_3UTT · 2023-10-25

**Soundness:** 2 fair
**Presentation:** 3 good
**Contribution:** 2 fair
**Rating:** 5
**Confidence:** 3

**Summary:**

This work shows that the loss of curvature is correlated with the loss of plasticity

**Strengths:**

The idea that the curvature relates to the plasticity phenomenon is novel and interesting. It is a plus because the curvature of the loss function is of central importance to deep learning theory. While I do like this work, I think there are aspects can be improved

**Weaknesses:**

There are a few problems that are probably fixable:

1. I find the current figures not sufficiently convincing. First of all, the srank is a normalized quantity, and therefore it cannot tell us the curvature of the loss. This is because the curvature is closely related to the absolute scale of the eigenvalues -- for a fixed sharpness of the local landscape, one can have an arbitrary srank and vice versa. Therefore, I am not convinced that the srank can be interpreted as "curvature"

2. The figure are not clearly convincing. In particular, what the authors are trying to argue is that the batch error increase is correlated with srank decrease -- but there is no single figure that precisely shows this. The authors need to include a figure whose x-axis is the srank and the y axis is the batch error, and perform a rigorous correlation analysis to convince me

3. I think it would be valuable to discuss the recent results in https://arxiv.org/abs/2309.16932 (mainly section 4.4). For example, the authors say that "This suggests that loss of palsticity only occurs with the non-linear activations," which does not seem true in light of the result in the related work. Also, throughout, the authors suggest that L2 regularization can possibly mitigate the loss of plasticity, whereas the results in the above work shows that L2 reg. exacerbates it. The authors should explain the contradictions and why they occur


Also, I have a question:
1. Why is the stable rank based on the singular values but not the eigenvalues? The singular values are only positive, but, in my opinion, the negative eigevalues in the Hesssian could play an important role here.

**Questions:**

See the weakness section

---

> ### Author Response · Authors · 2023-11-15
> **Thank you for your review!**
>
> Thank you for your review, below we address your specific concerns:
>
> >Definition of curvature, scale of eigenvalues and rank as a normalized quantity.
>
> Please see the shared reply, [Curvature Definitions], for more details
>
> There are several definitions of curvature, but the Hessian describes the local curvature of the function. The srank of the Hessian gives the number of curvature directions and we are arguing this property of the curvature explains loss of plasticity.
>
> The fact that the rank is a normalized quantity (normalized meaning it ignores the scale of the singular values) means that it provides information about the relative distribution of the singular values. Our findings suggest that this normalized quantity is useful for explaining plasticity.
>
> >include a figure whose x-axis is the srank and the y axis is the batch error
>
> Please see shared reply [Correlation Plots] and updated manuscript (Appendix A.1, D.3 and D.4) with correlation plots.
>
> >Discussing “Symmetry Leads to Structured Constraint of Learning”
>
> First, the paper that you link to was uploaded to arxiv after the ICLR submission deadline. We do appreciate the reference, as the paper looks quite interesting and relevant. Discussion of the linked paper, however, should not be relevant to the reviewing process as per the reviewer FAQ. That being said, we are not sure what conclusions to draw regarding linear networks in that paper. Furthermore, L2 regularization has been shown to mitigate plasticity loss in several other works, which we reference in Section 2.1, but it requires careful hyperparameter tuning.
>
> >Why is the stable rank based on the singular values but not the eigenvalues?
>
> The stable rank is always defined with respect to the singular values because the magnitude of the singular values is the magnitude of curvature in the corresponding. The stable rank measures the number of singular values that account for a large fraction of the sum of singular values.

---

> > ### Author Response · Authors · 2023-11-21
> > **Rebuttal Follow-up**
> >
> > Dear Reviewer,
> >
> > We are reaching out to see if our response addresses your concerns. As the discussion period is soon ending, we would greatly appreciate your confirmation that we have addressed your concerns.

---

### Official Review · Reviewer_iEeG · 2023-10-30

**Soundness:** 2 fair
**Presentation:** 3 good
**Contribution:** 2 fair
**Rating:** 5
**Confidence:** 4

**Summary:**

The paper proposes the rank of the network Hessian as an explanation of plasticity in neural networks, and conducts a series of empirical analyses to evaluate the utility of this explanation. It goes on to propose a regularizer which aims to minimize an approximation of the Wasserstein distance between the current distribution of per-layer weights and the initial distribution. This regularizer demonstrates improved stability and robustness to hyperparameters over related L2-based weight regularizers on the benchmarks studied.

**Strengths:**

- The paper makes a solid effort to find a quantifiable property of networks which lose plasticity without accumulating dead units. While Lyle et al. (2023) allude to changes in the structure of the loss landscape which can occur in networks which lose plasticity in the absence of dormant neurons, prior work has not identified a cheap scalar quantity which indicates that this pathological curvature has arisen.
- The paper evaluates its hypotheses and method on widely-used benchmarks in the study of plasticity loss.
- The idea of studying the *rank* of the Hessian, as opposed to its maximal eigenvalue, is interesting and not something I have seen explored much in the literature.
 - The paper provides a nice illustration of the limitations of update norm to explain plasticity loss, in contrast to the observation of Abbas et al. that reduction in update norm often correlated with plasticity loss in networks which accumulate dead ReLU units.
 - The paper also highlights the importance of taking into account differences in the initial value in a performance metric after a task change when evaluating learning algorithms. In particular, what I presume to be higher loss spikes due to increased confidence in the network's predictions cause methods which allow the network to more effectively minimize its loss on the prior task can lead to spurious large initial values in the loss after a task change. Evaluation schemes which take an average of the loss over the training trajectory then overestimate the loss in plasticity of these methods.
  - The study of the alignment between the Hessian principal components and the gradients could offer some potentially interesting insights into the evolution of the loss landscape.
  - The proposed regularizer is sensible, reasonably cheap (only introducing a log(n) factor on top of standard weight decay) and seems to be highly effective.

**Weaknesses:**

- The paper doesn't evaluate across different architectures, and only considers extremely small MLPs. Given that prior work has noted that an intervention's efficacy often heavily depends on the architecture to which it is applied, it is important to evaluate an intervention on a range of architectures to get a full view of its robustness.
- Saying that the curvature of the neural network is reduced because of a reduction in srank might go against existing notions of curvature in the literature on deep neural network training, which often characterize the curvature by the maximal Hessian eigenvalue. In particular, a reduction in srank of the features is often caused by the largest eigenvalue growing faster than the remainder. It is possible that a similar phenomenon is happening here, wherein the largest eigenvalue of the Hessian is growing faster than the other eigenvalues. A visualization of the eigenvalue distribution at the start vs at the end of training would be useful. I would be inclined to call a reduction in the srank an increase in the 'degeneracy' of the Hessian, rather than a reduction in curvature.
- The rank of the Hessian with respect to the last two layers of the network is likely to be highly correlated with the rank of the penultimate feature layer. Figures 1 and 2, for example, show that the rankings given by the srank of the Hessian and the srank of the features largely agree with each other. Similar issues in the correlation between the feature srank and the error over time also emerge in the Hessian srank, for example the leaky-ReLU network in Figure 2, middle column. The evaluation of the srank of the features in Figure 1 appears to have either been performed with a very narrow network or a very small batch size, making it difficult to evaluate the significance of these measurements. I think the paper would benefit from a closer examination of the difference between the Hessian rank and the feature rank in these networks.
- One of the biggest issues that emerges from the paper is the observation that the Hessian srank often exhibits similar inconsistencies to other quantities, like the srank of the features, which were used as justification to dismiss these quantities as good explanations of plasticity. For example, weight decay and regenerative regularization both exhibit similar Hessian srank, but weight decay exhibits greater plasticity loss in Figure 4.
- The choice to study the rank of the Hessian, as opposed to the rank of the empirical NTK or the features, is not given sufficient motivation. Each of these quantities reveals something meaningful about the loss landscape, and it is not immediately obvious that the Hessian should be preferred over the other two.
- While the proposed regularizer is reasonable, it feels very much orthogonal to the curvature of the network. The causal connection between preserving the distribution of the parameters and maintaining the rank of the hessian remains vague, and the benefits of the regularizer on the rank appear incidental.
- Because of the limited set of evaluations, it is difficult to say whether the proposed regularizer will be useful in other contexts, where it may be necessary for the distribution of weights to shift in ways that increase the Wasserstein distance from initialization, or where the regularizer may slow down learning too much to be practical. It would be useful to evaluate on other datasets and to also show the learning curves; for example, the worse online learning results for the wasserstein regularization coupled with the better end-of-training results suggest that not only does the loss start out much higher, but that it stays higher than the other methods for a long time -- in other words, it is resulting in much slower learning.

**Questions:**

- I am skeptical of the results which seem to indicate that a linear network is able to attain near-0 error on MNIST. I have trained linear classifiers on this dataset in the past and don't recall seeing anything exceed 80-90% accuracy. I am also extremely surprised at the results which seem to indicate that a 3-layer, < 128-width **linear** MLP is able to attain zero training error on label-shuffled CIFAR-10. For example, even the larger networks studied by Zhang et al. in the classic paper "Understanding deep learning requires rethinking generalization" required dozens to hundreds of epochs to converge on random CIFAR-10 labels. I have a difficult time believing that a task sufficiently difficult for an ImageNet-calibre architecture to require Could the authors confirm that they are indeed able to obtain zero error on image classification datasets with a linear model?
- How strong are the correlations between feature, gradient, and Hessian srank, and can the authors provide evidence that the Hessian srank is providing a meaningful notion of curvature or trainability not captured by the other two quantities?
- The paper would benefit from evaluations in other architectures, and on tasks that cannot be solved by a linear model, to validate the utility of the proposed regularizer. In particular, how does the regularization method perform on tasks where linear networks fail? Does it generalize to convolutional networks / ResNets / sequence models?
- Can the authors explain the settings noted in the "weaknesses" section where curvature appears to fail to explain plasticity in neural networks?

---

> ### Author Response · Authors · 2023-11-15
> **Thank you for your review!**
>
> Thank you for your review, below we address your specific concerns:
>
>
> >a linear network is able to attain near-0 error on MNIST(?)
>
> Our experiments use a subset of the MNIST dataset (similar to [1,2]), and a linear network can achieve a low error on this subset. The non-linear networks achieve a lower error than the linear network on the first task (see Appendix D.5), but due to loss of plasticity this error increases to be much higher than the error of the linear network. We emphasize that it is striking that loss of plasticity still occurs on problems for which linear features can be predictive.
>
> >correlations between feature, gradient, and Hessian srank, and can the authors provide evidence that the Hessian srank is providing a meaningful notion of curvature or trainability not captured by the other two quantities?
>
> Please see shared reply and updated manuscript (Appendix A.1, D.3 and D.4). These results validate the fact that the number of curvature directions is important in explaining plasticity.
>
> >The paper doesn't evaluate across different architectures, and only considers extremely small MLPs. [...] The paper would benefit from evaluations in other architectures … Does it generalize to convolutional networks / ResNets / sequence models?
>
> Please see the shared reply [Neural Network Architecture] and updated manuscript (Appendix D.2, D.4)
> We speculate that a resnet would mitigate some plasticity loss because its residual connections provide a deep linear subnetwork, but we leave this for future work.
>
> >reduction in srank might go against existing notions of curvature in the literature on deep neural network training, which often characterize the curvature by the maximal Hessian eigenvalue.
>
> See shared reply, under [Curvature Definitions]
>
> >I think the paper would benefit from a closer examination of the difference between the Hessian rank and the feature rank in these networks.
>
> The Hessian is a second-order property of the objective function, describing the local curvature and its rank counts the number of directions of curvature. In contrast, the feature representation is a zeroth-order property of the neural network. For example, if the labels are shuffled between tasks then the feature rank at the end of one task and at the beginning of the subsequent task would be the same. Thus, the feature rank cannot distinguish between some changes in the optimization landscape. There is no a-priori reason to believe that the feature rank is related to curvature or the Hessian rank, and indeed previous papers have also found that feature rank is not a consistent explanation for plasticity loss [2].
>
> >For example, weight decay and regenerative regularization both exhibit similar Hessian srank, but weight decay exhibits greater plasticity loss in Figure 4.
>
> Although the Hessian srank decreases in the regularized experiment, the rank is still substantially larger than the unregularized setting, even compared to the linear baseline. We do not claim that weight decay exhibits plasticity loss in Figure 4, as the confidence interval is large we conclude that weight decay is unstable. See the new correlation plots for further evidence.
>
> >The choice to study the rank of the Hessian, as opposed to the rank of the empirical NTK or the features, is not given sufficient motivation.
>
> The motivation is twofold. First, previously proposed explanations of plasticity have focused on zeroth (feature rank) and first (gradient/update norm) order properties. Second, there is a growing literature finding that curvature (which the Hessian describes locally) strongly influences learning dynamics. It is thus a natural hypothesis that the Hessian may help explain loss of plasticity.
>
> >While the proposed regularizer is reasonable, it feels very much orthogonal to the curvature of the network.
>
> Please see shared reply, [Connecting Curvature and Regularization]
>
>
> > for example, the worse online learning results for the wasserstein regularization coupled with the better end-of-training results suggest that not only does the loss start out much higher, but that it stays higher than the other methods for a long time -- in other words, it is resulting in much slower learning.
>
> We found that our proposed Wasserstein regularizer does not result in slower learning than the regen regularizer (See Appendix D.5 for inter-task learning curves). In general, the Wasserstein regularizer is smaller than the L2 and regen regularizer. Thus, there is a larger set of parameters within a set distance with respect to the Wasserstein distance.
>
> [1] Kumar et al. Maintaining Plasticity via Regenerative Regularization, CoRR, abs/2308.11958v1, 2023
> [2] Lyle et al. Understanding Plasticity in Neural Networks. ICML 2023

---

> ### Comment · Reviewer_iEeG · 2023-11-19
>
> Thanks to the authors for their response, which has clarified a number of my questions. However, several concerns remain.
>
> 1. **Success of identity activation on MNIST:** can the authors specify the exact number of samples used and include this in the experiment description? I am skeptical of the significance of results on a task which can be solved by a linear model.
> 2. **Network architecture:** thanks to the authors for clarifying that a convolutional network is used in the CIFAR-10 experiments. However, in light of the above response I do need to express my concern over the significance of including different architectures given that the task can still be solved perfectly by a linear model.
> 3. **Hessian vs feature rank:** I am not convinced by the authors’ response that “there is no a-priori reason to believe that the feature rank is related to curvature or the Hessian rank, and indeed previous papers have also found that feature rank is not a consistent explanation for plasticity loss.” Given that the Hessian is the ‘derivative’ of the gradient, and that gradient geometry is closely connected with feature geometry, it is intuitive that the rank of the features would give a rough sense of the degeneracy of the Hessian, particularly if considered over the course of training on a few structurally similar tasks. Further, the similar trends exhibited by Hessian rank and feature rank suggest a connection between the two. I would be more convinced of the independence of these two quantities if they made different predictions about which networks are likely to lose plasticity or could be shown to have differing trends in different experimental settings.
> 4. **Justification of title:** I remain unconvinced that the paper’s titular claim, that “curvature explains plasticity” is supported by the experiments provided. The experimental results in the paper show that the Hessian rank does often correlate with plasticity, but so do other quantities which were shown by Lyle et al. to not fully explain plasticity. Further, there are examples in the paper itself which “falsify” Hessian rank as an explanation, for example Figure 3, where batch error and Hessian rank both increase after the 20th task.

---

> > ### Author Response · Authors · 2023-11-21
> > **Thanks for your reply!**
> >
> > Thank you for your reply and engagement, we appreciate the feedback. We have updated the manuscript with an additional result on a larger subset of MNIST (Appendix E). We provide details below while we address your questions:
> >
> > >Success of identity activation on MNIST
> >
> > The MNIST experiment uses 1280 samples (see Appendix B.1). We understand that a linear model performing well on this problem is strange given that MNIST is a high-dimensional classification problem, even given that nonlinear models lose plasticity. Indeed a linear model has an effective 784*10 parameter count, which allows it to fit 1280 samples. We have included experiments with a subset larger than the effective linear parameter count (12800, see appendix E).
> >
> > The results in this experiment are similar but more pronounced than in the smaller subset of MNIST. The linear model does not lose plasticity but it also can no longer achieve low error. The non-linear activation functions can achieve a much lower error in the first task but, due to loss of plasticity, eventually becomes worse than the linear model. Regularization mitigates loss of plasticity and prevents reduce in the Hessian sRank, contributing further evidence that loss of plasticity is well explained by the Hessian sRank.
> >
> > >Hessian vs feature rank
> >
> > You are correct to state that the feature representation appears in the gradient (in particular, for the gradient of the last layer’s weights). But, the feature representation does not carry information about curvature, nor its rank. There are cases in which the model has a high feature rank and low hessian rank (see LayerNorm line, top and bottom right plot in Figure 26, Appendix E). There are also cases in which we can have a constant Hessian Rank and decreasing feature rank (see Wasserstein regularization line, top and bottom right plot in Figure 26 in Appendix E).
> >
> > > Justification of title
> >
> > We would like to emphasize that the connection between curvature and plasticity has been previously noted by Lyle et al. (2023), and that there is a growing literature that curvature is important for understanding neural network training dynamics. Our paper provides substantial evidence to support the claim that curvature explains loss of plasticity, while also corroborating the insufficiency of other explanations.
> >
> > Thank you for noticing the detail in Figure 3, but that Hessian srank is for the task-end. Our explanation for plasticity loss is with respect to the task-beginning Hessian srank, which is not depicted in Figure 3. The task-end Hessian srank depicted in Figure 3 is often small, but this is expected because the task local optimum is flat in several directions. The Hessian srank at the task beginning should not be small, because the model will need to learn from experience in the new task and it cannot easily learn if the optimization landscape is flat in several directions.

---

### Official Review · Reviewer_BAzs · 2023-10-31

**Soundness:** 3 good
**Presentation:** 4 excellent
**Contribution:** 3 good
**Rating:** 6
**Confidence:** 4

**Summary:**

This paper investigates the loss of plasticity in learning algorithms, where they struggle to adapt to changing environments. While previous research linked this to factors like optimizer type, gradient norm, and neuron dormancy, this study introduces a novel perspective: loss of plasticity is tied to a decrease in curvature. Specifically, the authors highlight a correlation between the loss of plasticity and the reduced rank of the Hessian of the training objective at new task commencements. To combat this, the paper introduces a regularizer that retains weight distributions close to their initial setups, preserving curvature and ensuring better adaptability across successive tasks.

**Strengths:**

This paper offers a novel perspective on the loss of plasticity by associating it with curvature in the optimization landscape. This approach deepens our understanding of plasticity and distinguishes this work from previous research.

At the core of the paper's approach is an examination of adaptability in curvature using the rank of the Hessian matrix. This focus on how task alterations affect curvature offers a detailed and precise metric, setting this work apart from broader methodologies in earlier studies. Moreover, the authors complement theoretical discussions with empirical evidence, bolstering the paper's credibility and relevance. While past research has suggested various metrics such as neuron dormancy, decrease in active units, feature rank, and weight norm to measure plasticity, the authors demonstrate that these metrics do not consistently track the loss of plasticity.

Furthermore, this paper proposes a Wasserstein initialization regularizer. This not only mitigates the issues identified but also paves the way for the adaptability of learning algorithms.

**Weaknesses:**

This paper contains interesting ideas and promising experimental results. However, I do have several concerns regarding the completeness of this paper to comprehensively support their claims. Specifically:

Metrics-Related Concerns:
- The authors' decision to measure the partial blockwise of the last two layers raises questions. It's unclear why early layers are seemingly disregarded. Don't they play a significant role in the loss of plasticity?
- For a comprehensive evaluation, it would be valuable if the authors could provide a layer-wise or partial block-wise Hessian rank for the experiments. Especially, I would like to check it out for the CIFAR-10 experiments, since it utilizes a deeper network than Mnist experiments.

Methodological Concerns:
- The choice of the Wasserstein distance measure as a regularizer adds complexity, especially when it comes to tuning for smaller sample sizes ($d$). Why not employ the simpler L2 regularization from the initial parameter? Does the norm of the neural network should be large enough to make accurate predictions for a large number of tasks?
- It remains ambiguous how the Wasserstein distance regularizer directly contributes to increasing the rank of the Hessian matrix. The inclusion of theoretical evidence or explanations that tie these concepts together would strengthen the paper's methodological claims.

Experimental Concerns:
- The omission of experiments using permuted CIFAR-10 seems to be an oversight, given its relevance in evaluating plasticity.
- A more rigorous comparison with related works is needed:
  - A comparison with widely used techniques like dropout and data augmentation would be informative.
  - A comparison w.r.t plasticity literature including CReLU activation [1], Layer Normalization [2], Last layer reset [3], or even combinations of these approaches [4].

[1] Loss of Plasticity in Continual Deep Reinforcement Learning., Abbas et al., TMLR 2023.

[2] Understanding plasticity in neural networks., Lyle et al., ICML 2023.

[3] Primacy Bias in Deep Reinforcement Learning., Nikishin et al., ICML 2022.

[4] Enhancing input and label plasticity for Sample Efficient Reinforcement Learning., Lee et al., NeurIPS 2023.

I'm willing to increase the score if the authors address the outlined limitations.

**Questions:**

N/A

---

> ### Author Response · Authors · 2023-11-15
> **Thank you for your review!**
>
> Thank you for your review, below we address your specific concerns:
>
> >It's unclear why early layers are seemingly disregarded.
>
> The goal of this paper is to compute, exactly, the Hessian rank since the main claim of our paper is that the curvature, as measured by the Hessian rank, explains loss of plasticity. This computation has a O(d^3) cost where d is the number of parameters. It turns out that we do not need the exact hessian, because layer-wise approximations provide a good approximation of the Hessian for stationary tasks [1].
>
> > provide a layer-wise or partial block-wise Hessian rank
>
> While we have argued that the layer-wise Hessian is a good approximation for stationary tasks, it is not a good approximation for non-stationary tasks like in our continual learning experiments.
> This is because the layer-wise Hessian does not change when the target function changes if we are using a piecewise-linear activation function like ReLU.
> Taking the last 2 layers is the simplest way to avoid this issue with the layerwise approximation, and our results demonstrate that it is sufficient to explain plasticity loss.
>
> >The choice of the Wasserstein distance measure as a regularizer adds complexity, especially when it comes to tuning for smaller sample sizes  Why not employ the simpler L2 regularization from the initial parameter?
>
> The Wasserstein distance is not any more complicated than the L2 or regen (which is L2 regularization from the initial parameter). Like both L2 and regen, it uses a single hyperparameter and does not depend on the sample size. In Appendix C.5, we show that the Wasserstein regularizer is actually more robust to the hyperparameter than both L2 and regen.
>
> >It remains ambiguous how the Wasserstein distance regularizer directly contributes to increasing the rank of the Hessian matrix.
>
> See shared reply [Connecting Curvature and Regularization]
>
> >The omission of experiments using permuted CIFAR-10 seems to be an oversight
>
> Thank you for bringing this up as it was not made explicit in the manuscript.
> Observation permuted CIFAR 10 is not a common benchmark for studying plasticity, because the CIFAR experiment uses a convolutional neural network. Observation permutations would destroy spatial regularity, and the translation invariance of convolutions would be an improper inductive bias.
>
> >Comparison with dropout, CReLU, layer norm, last layer reset
>
> Thank you for your suggestions, please see the shared reply [Additional Experiments] and updated manuscript (Appendix D.2, D.3 and D.4)
>
> [1] Sankar et al. A deeper look at the hessian eigenspectrum of deep neural networks and its applications to regularization, AAAI 2021

---

> > ### Comment · Reviewer_BAzs · 2023-11-21
> >
> > Thank you for addressing the previous concerns and for including comparisons with ReDO, CReLU, and Reset in your additional experiments. These comparisons significantly contribute to contextualizing your findings within the broader spectrum of existing methodologies.
> >
> > However, I still have a critical concern regarding the **metric** used for analyzing plasticity:
> >
> > The paper claims that measuring the Hessian rank of the last two layers is sufficient to analyze the network's loss of plasticity. While this approach has merits, its scalability and depth-specific applicability raise questions. In models with a large number of layers, it's unclear whether focusing solely on the last two layers adequately captures the network's overall plasticity. A more comprehensive layer-wise analysis could offer deeper insights into where and how plasticity loss occurs throughout the network. Additionally, considering the limitations posed by ReLU activation in layer-wise analysis, exploring two-layer-wise analysis could provide a more nuanced understanding of the phenomenon across different network depths.
> >
> > Still, I believe this work provides good insights and am inclined to raise my score if this concern can be relieved.

---

> > > ### Author Response · Authors · 2023-11-22
> > >
> > > Thank you for your reply. You express some concern whether the last two layers can accurately capture curvature for deeper networks. We have added results using a 13-layer neural network in Appendix F. Our results show that, with this 13 layer neural network, the sRank of the Hessian with respect to the last two layers exhibits the same trend with plasticity loss, thereby validating its use as an approximation. For further discussion on why the last 2 layers are effective, please see [Previous Layers].
> > >
> > > You also raise a very interesting question: 1) can the Hessian sRank help identify and localize loss of plasticity to particular layers or regions of the network?
> > >
> > > [Previous Layers] We first note that the previous layers still contribute to the input for the last 2 layers, which can influence the Hessian srank. This is why, for example, the Hessian sRank decreases even when resetting the last 2 layers (see Figure 20). Although the Hessian is taken with respect to the last two resetted layers, the outputs of the previous layers are used as input for the last 2 layers and drive the reduction in the srank.
> > >
> > > This provides further evidence that the Hessian-based metrics we analyze provide a good surrogate for the curvature of the whole neural network. It is not clear, however, how this can be used to localize and identify particular weights with low plasticity. While your suggestion is very interesting as a potential pruning strategy, we leave this for future work.

---

> > > > ### Comment · Reviewer_BAzs · 2023-11-23
> > > >
> > > > I appreciate the additional results in Appendix F addressing my concerns about the capability of the last two layers to capture curvature in deeper networks. The correlation between the sRank of the Hessian and plasticity loss in a 13-layer neural network, as discussed in [Previous Layers], convincingly validates the approach. While the potential of Hessian sRank to localize plasticity loss within specific network regions is intriguing, I understand this remains an area for future research. The response reinforces my confidence in the value of your work, and **I am increasing my score to 6**.
> > > >
> > > > As a side note, I have a question driven by curiosity rather than a critique of the work's current scope. How would this framework perform if applied to more common architectures, such as ResNet18 or VGG16, particularly in the context of the CIFAR-10 dataset? Specifically, I think the readers will be interested in whether the phenomenon of loss of plasticity is still observable under these conditions and if Wasserstein regularization continues to be effective in this scenario.

---

### Official Review · Reviewer_sWyZ · 2023-11-01

**Soundness:** 3 good
**Presentation:** 3 good
**Contribution:** 3 good
**Rating:** 6
**Confidence:** 4

**Summary:**

This paper posits a novel explanation for the loss of plasticity in neural networks, attributing it to the loss of curvature. The authors challenge existing explanations for the plasticity loss by providing counter examples and highlighting their inconsistencies. The plasticity loss, which refers to the phenomenon of neural networks losing their ability to learn from new experience, can be observed by increase in training error as the network learns. To measure the curvature, this paper exploited a partial blockwise neural network. Moreover, this paper suggests that a Wasserstein regularizer, which keeps the distribution of parameters close to their initialization, prevents the loss of plasticity.

**Strengths:**

- The paper is addressing important problem. Motivation describing the inconsistency of existing explanations was straightforward.

- The paper is clearly written and well structured. It was easy to follow and understand the authors’ point.

- It is interesting that the Wasserstein regularization effectively prevents the loss of plasticity.

**Weaknesses:**

[Significance of results] The presented arguments that existing explanations for the loss of plasticity are inconsistent seems to be insufficient. Although the proposed explanation related to the loss of curvature is intriguing, the authors should provide additional experiments (diverse model architectures and benchmarks) to argue that the proposed one is a consistent explanation for the plasticity loss. Moreover, it would be great if the authors could measure the consistency quantitatively.

[Evaluation metric in continual learning setup] The paper seems to overlook an analysis on forgetting which is one of the most important criteria in the continual learning setup. It is recommended that the authors expand their discussions to encompass the concept of forgetting, in addition to delving deeper into plasticity.

[Minor typo] In Appendix A bullet points 3 and 4, there are some missing symbols between two numbers (e.g., at the level of 2.6 2.7).

**Questions:**

As mentioned in the weakness section, the significance of inconsistency in existing explanations is unclear. In Section 3 results, the rank of representation (top right of Figure 1) decreased, and the batch error increased when using tanh. As tanh is experiencing small decrease on the rank of representation, the batch error also showed relatively small increase. This seems to be a consistent explanation for the loss of plasticity. More explanations on this will strengthen the authors’ argument.

In addition, why is Wasserstein regularization better than regenerative regularization? As mentioned in Section 5, the order statistics is defined as the difference between two regularizations. It would be beneficial for the readers if the authors can provide explanations regarding the substantial difference in the results presented in Figure 5.

---

> ### Author Response · Authors · 2023-11-15
> **Thank you for your review!**
>
> Thank you for your review, below we address your specific concerns:
>
> > the authors should provide additional experiments … Moreover, it would be great if the authors could measure the consistency quantitatively.
>
> Please see the shared reply [Additional Experiments] and updated manuscript (Appendix A.1, D.3 and D.4).
>
> >Forgetting
>
> Our work focuses on the problem of plasticity, and is explicitly isolated from the problem of forgetting. Our experiments, for example, measure error with respect to the task being trained and not on any previous tasks. This is by design; we want to cultivate a better understanding of continual learning by first explaining the phenomenon of plasticity.
>
> > As tanh is experiencing small decrease on the rank of representation, the batch error also showed relatively small increase.
>
> The feature rank is an inconsistent explanation because, for tanh, the increase in the batch error occurs and continues when the feature rank is constant. We have added correlation plots to show this more clearly, please see the shared reply [Correlation Plots] and the updated manuscript (Appendix A.1)
>
> >why is Wasserstein regularization better than regenerative regularization?
>
> Please see shared reply, [Connecting Curvature and Regularization]

---

> > ### Author Response · Authors · 2023-11-21
> > **Rebuttal Follow-up**
> >
> > Dear Reviewer,
> >
> > We are reaching out to see if our response addresses your concerns. As the discussion period is soon ending, we would greatly appreciate your confirmation that we have addressed your concerns.

---

> > ### Comment · Reviewer_sWyZ · 2023-11-23
> >
> > Thank you to the authors for their efforts in addressing the concerns previously raised regarding forgetting and regularization. I recommend incorporating additional details in the final manuscript to provide more comprehensive explanations.
> >
> > While I acknowledge the improved explanation of plasticity through Hessian rank, as presented by the authors, I remain partially unconvinced about the overarching claim linking the loss of plasticity solely to Hessian rank. For instance, in Figure 7 (top left), the correlation plot involving identity activation is obscured by other elements, casting ambiguity on the methodology used for drawing the fitted line. Furthermore, it would be beneficial for the authors to offer explicit guidance on interpreting the correlation plots. Specifically, delineating which aspects yield consistent explanations and which areas present contradictory evidence would greatly enhance the clarity and usefulness of these visual representations.

---

### Author Response · Authors · 2023-11-15
**Shared reply**

We thank the reviewers for their feedback. In this shared reply, we address the concerns that were mentioned by more than one reviewer, and we address specific concerns in the individual replies.

We were pleased to see that the paper does a good job communicating its central goal and that all reviewers thought the proposed regularizer to be both novel and interesting. To help retrieve the context of our paper, and to ground the discussion, we want to reiterate the goal of the paper is to propose and to evaluate curvature as an explanation for loss of plasticity. We specifically posit that a reduction in the number of curvature directions, as measured by a rank of the Hessian, explains loss of plasticity.

## Concerns Raised, and Changes Made:

- [Additional Experiments] A common concern regarding our paper was the datasets we used in our evaluation. We want to emphasize that this choice is relatively standard, as several continual learning problems study MNIST and CIFAR with non-stationarity [2,4,5]. Nevertheless, after the reviewers’ comments we were curious whether the results scale to a more challenging benchmark and we ran it on the relatively new and large scale Continual Imagenet (proposed in [3], also used in [4]). The results are now in Appendix D and include comparison against other methods that purport to mitigate plasticity loss: layernorm, dropout, CReLU, and a last layer reset (as suggested by Reviewer BAzs). We are happy to confirm that these new results also fully support the claims we make in the paper.

- [Neural Network Architecture] We also realized that there was a misunderstanding on the types of models we evaluated, as Reviewers iEeG and sWyZ pointed out for the need to study larger models than MLPs. Fortunately, the results presented on CIFAR in Figure 2(right) and Figure 6 are using convolutional neural networks, as well as the results on Continual Imagenet in Appendix D. As suggested by Reviewer BAzs, we have also included different neural network modifications in Appendix D.2.

- [Correlation Plots] Reviewers sWyZ, iEeG, and 3UTT noted that the plots presented in Section 3 are difficult to understand and we have added correlation plots to show more clearly the correlation. The inconsistency of previous explanations and the consistency of the Hessian sRank is demonstrated in Appendix A.1 and further correlation studies of the curvature are conducted on Continual Imagenet in D.3 and D.4.

There were two points of confusion that we would like to make clear:

## [Curvature Definitions]
First, some reviewers expressed confusion around the term “curvature”.
The reviewers are correct that there are several terms related to curvature in the literature. Specifically, the maximum singular value of the Hessian, or sharpness [1], is a common measure of curvature. Our results in Appendix D.4 show that sharpness is also not consistent with loss of plasticity.

The rank of the Hessian counts the number of directions of curvature, and so it is a measure of curvature as well. We have shown that it is a consistent explanation for plasticity loss. Thus, the central claim that curvature explains loss of plasticity is factual, and reflective of our contribution.

## [Connecting Curvature and Regularization, Motivating Wasserstein]
Second, there was confusion regarding the connection between regularization and curvature. Because the regularizers we study are twice-differentiable, they have a direct effect on the Hessian and, thus, curvature. Regularization is often used in classical regression to increase the rank of the Hessian and it has been shown to be a simple, yet effective, approach to maintaining plasticity in previous work.

  The motivation for the Wasserstein regularizer is that both L2 and regen can be sensitive to its hyperparameter, suggesting that they can easily overregularize. The distributional distance used in the Wasserstein distance is "looser" than L2 or regen, and thus can be potentially more stable and allow for a larger set of possible parameter values. To see this, note that for a specific parameter and initialization, we know that Wass < regen  (the difference between two sorted lists is always less than two unsorted lists). Generally, it is the case that regen < L2. Thus, the benefit of our Wasserstein regularizer is that it is “looser” and adds a small perturbation to the curvature of the landscape relative to the other regularizers.

## References
[1] Cohen et al. Gradient Descent on Neural Networks Typically Occurs at the Edge of Stability, ICLR 2021
[2] Dohare et al. Continual Backprop: Stochastic Gradient Descent with Persistent Randomness, CoRR, abs/2108.06325v3, 2021
[3] Dohare et al. Maintaining Plasticity in Deep Continual Learning, CoRR, abs/2306.13812, 2023
[4] Kumar et al. Maintaining Plasticity via Regenerative Regularization, CoRR, abs/2308.11958v1, 2023
[5] Lyle et al. Understanding Plasticity in Neural Networks. ICML 2023

---

### Meta-Review · Area_Chair_FPDe · 2023-12-02

**Metareview:**

One issue in continual learning is that neural networks lose their ability to learn from new tasks, which is called loss of plasticity in this paper. Several explanations exist, but the authors give counterexamples to these previous explanations and provide a new angle from the negative curvature directions of Hessian, measured by the effective rank of Hessian with respect to weights. While the paper gave some correlation analysis between the curvature information and plasticity (e.g., Figure 7), reviewers find the evidence not supportive enough of the claimed explanation.

**Justification For Why Not Higher Score:**

The paper can be improved in several aspects.
- Some of the choices in the methods seem arbitrary. For example, while the paper talks about curvature, the method specifically uses the effective rank of the Hessian with respect to the last two layers' parameters.
- I agree with reviewer 3UTT that the definition of stable rank is strange here. As far as I know, the definition used in this paper is more like an effective rank. For example, the authors can check section 7.6.1 in "High-Dimensional Probability - An Introduction with Applications in Data Science", Vershynin 2020.
- The results are obtained for relatively small-scale networks. Experimental validations on larger scales are useful in improving the impact of this research.
- While the main message in Figure 7 seems clear, the purple curve is covered by other curves in Figure 7 (top left). Better interpretations of the Figure are necessary.

**Justification For Why Not Lower Score:**

N/A

---

### Decision · Program_Chairs · 2024-01-16

Reject